# Impact of anthropogenic emissions on biogenic secondary organic aerosol: Observation in the Pearl River Delta, South China

Yu-Qing Zhang[1, *], Duo-Hong Chen[2, *], Xiang Ding[1, †], Jun Li[1], Tao Zhang[2], Jun-Qi Wang[1], Qian Cheng[1], Hao Jiang[1], Wei Song[1], Yu-Bo Ou[2], Peng-Lin Ye[3], Gan Zhang[1], Xin-Ming Wang[1, 4]

State Key Laboratory of Organic Geochemistry and Guangdong Provincial Key Laboratory of Environmental Protection and Resources Utilization, Guangzhou Institute of Geochemistry, Chinese Academy of Sciences, Guangzhou, 510640, China
State Environmental Protection Key Laboratory of Regional Air Quality Monitoring, Environmental Monitoring Center of Guangdong Province, Guangzhou, 510308, China
Aerodyne Research Inc., Billerica, Massachusetts 01821, United States
Center for Excellence in Regional Atmospheric Environment, Institute of Urban Environment, Chinese Academy of Sciences, Xiamen, 361021, China

* These authors contributed equally to this work.

† *Correspondence to*: Xiang Ding (xiangd@gig.ac.cn)

**Abstract.** Secondary organic aerosol (SOA) formation from biogenic precursors is affected by anthropogenic emissions, which is not well understood in polluted areas. In the study, we accomplished a year-round campaign at nine sites in the polluted areas located in Pearl River Delta (PRD) region during 2015. We measured typical biogenic SOA (BSOA) tracers from isoprene, monoterpenes, and β-caryophyllene as well as major gaseous and particulate pollutants and investigated the impact of anthropogenic pollutants on BSOA formation. The concentrations of BSOA tracers were in the range of 45.4 to 109 ng m$^{-3}$ with the majority composed of products from monoterpenes (SOA$_M$, 47.2 ± 9.29 ng m$^{-3}$), followed by isoprene (SOA$_I$, 23.1 ± 10.8 ng m$^{-3}$), and β-caryophyllene (SOA$_C$, 3.85 ± 1.75 ng m$^{-3}$). We found that atmospheric oxidants, O$_x$ (O$_3$ plus NO$_2$), and sulfate correlated well with later-generation SOA$_M$ tracers, but not so for first-generation SOA$_M$ products. This suggested that high O$_x$ and sulfate could promote the formation of later-generation SOA$_M$ products, which probably led to relatively aged SOA$_M$ we observed in the PRD. For the SOA$_I$ tracers, not only 2-methylglyceric acid (NO/NO$_2$-channel product), but also the ratio of 2-methylglyceric acid to 2-methyltetrols (HO$_2$-channel products) exhibit NO$_x$ dependence, indicating the significant impact of NO$_x$ on SOA$_I$ formation pathways. The SOA$_C$ tracer elevated in winter at all sites and positively correlated with levoglucosan, O$_x$, and sulfate. Thus, the unexpected increase of SOA$_C$ in wintertime might be highly associated with the enhancement of biomass burning, O$_3$ chemistry and sulfate component in the PRD. The BSOAs that were estimated by the SOA tracer approach showed the highest concentration in fall and the lowest concentration in spring with an annual average concentration of 1.68 ± 0.40 μg m$^{-3}$. SOA$_M$ dominated the BSOA mass all year round. We also found that BSOA correlated well with sulfate and O$_x$. This implicated the significant effects of anthropogenic pollutants on BSOA formation and highlighted that we could reduce the BSOA through controlling on the anthropogenic emissions of sulfate and O$_x$ precursors in polluted regions.

## 1 Introduction

Secondary organic aerosols (SOA) that are produced through homogenous and heterogeneous processes of volatile organic compounds (VOCs) have significant effects on global climate change and regional air quality (von Schneidemesser et al., 2015). Globally, the emissions of biogenic VOCs (BVOCs) are dominant over anthropogenic VOCs. Thus, biogenic SOA (BSOA) is predominant over anthropogenic SOA. In the past decade, laboratorial, field, and modeling studies have demonstrated that BSOA

formation is highly affected by anthropogenic emissions (Zhang et al., 2015; Hoyle et al., 2011; Carlton et al., 2010). Increasing $NO_x$ shifts isoprene oxidation from the low-$NO_x$ conditions to the high-$NO_x$ conditions (Surratt et al., 2010) and enhances nighttime SOA formation via nitrate radical oxidation of monoterpenes (Xu et al., 2015). High $SO_2$ emission leads to abundant sulfate and acidic particles, which accelerates the BSOA production by the salting-in effect and acid-catalyzed reactions (Offenberg et al., 2009; Xu et al., 2016). In polluted regions, the increase of $O_3$ levels due to high emissions of $NO_x$ and VOCs, likely results in significant SOA formation through the ozonolysis of BVOCs (Sipilä et al., 2014; Riva et al., 2017). In addition, large emission and formation of anthropogenic organic matter (OM) in urban areas enhance the incorporation of BVOCs' oxidation products into the condensed phase (Donahue et al., 2006). Recently, Carlton et al. (2018) found that the removal of anthropogenic emissions of $NO_x$, $SO_2$, and primary OA in the CMAQ simulations could reduce BSOA by 23, 14, and 8% in summertime, respectively.

The Pearl River Delta region (PRD) (Figure 1a) is the most developed region in China. Rapid economic growth during the past three decades has resulted in large amounts of anthropogenic emissions in the PRD (Lu et al., 2013). Our observation during fall-winter season in 2008 at a regional site of the PRD showed that daily $PM_{2.5}$ was as high as 150 μg m$^{-3}$ (Ding et al., 2012). Fortunately, due to more and more strict and effective pollution controls in the PRD, $PM_{2.5}$ concentrations have significantly shrunk during the last decade and met the national ambient air quality standard (NAAQS) for annual-mean $PM_{2.5}$ (35 μg m$^{-3}$) since the year of 2015 (Figure 1b). However, $O_3$ and oxidant ($O_x$, $O_x = O_3 + NO_2$) are still in high levels and do not decrease apparently (Figure 1b). Hofzumahaus et al., (2009) observed extremely high OH concentrations in the PRD and proposed a recycling mechanism which increases the stability of OH in the air of polluted regions. All these indicate high atmospheric oxidative capacity in the PRD, since $O_3$, $NO_x$ and OH are intimately linked in atmospheric chemistry. On the other hand, BVOCs emissions in the PRD are expected to be high all the year in such a subtropical area (Zheng et al., 2010). In the process of such a dramatic change in air pollution characteristics (e.g. $PM_{2.5}$ and $O_3$), BSOA origins and formation mechanisms in the PRD should be profoundly affected in the last decade. In this study, year-round $PM_{2.5}$ samples were collected at nine sites in the PRD during 2015. We investigated SOA tracers from typical BVOCs (isoprene, monoterpenes, and β-caryophyllene) across the PRD for the first time. We checked seasonal variations in concentrations and compositions of these BSOA tracers and evaluated the impact of anthropogenic pollutants on BSOAs formation in the PRD. We also accessed the

SOA origins and discussed the implication in further reducing BSOA through controlling on the anthropogenic emissions.

## 2 Experimental Section

### 2.1 Field Sampling

Concurrent sampling was performed at 9 out of 23 sites in the Guangdong-Hong Kong-Macao regional air quality monitoring network (http://www.gdep.gov.cn/hjjce/, Figure 1a), including three urban sites in Zhaoqing (ZQ), Guangzhou (GZ) and Dongguan (DG), two suburban sites in Nansha (NS) and Zhuhai (ZH), and four rural sites in Tianhu (TH), Boluo (BL), Heshan (HS) and Taishan (TS).

At each site, 24-hr sampling was conducted every six days from January to December in 2015 using a $PM_{2.5}$ sampler equipped with quartz filters (8 × 10 inches) at a flow rate of 1.1 $m^3$ $min^{-1}$. Additionally, field blanks were collected monthly at all sites. Blank filters were covered with aluminum foil and baked at 500 ºC for 12 hrs and stored in a container with silica gel. After sampling, the filter samples were stored at −20 ºC.

In this study, the filters collected in January, April, July and October 2015 were selected to represent winter, spring, summer, and fall samples, respectively. A total of 170 field samples (4-5 samples for each season at each site) were analyzed in the current study.

### 2.2 Chemical Analysis

For each filter, organic carbon (OC) and elemental carbon (EC) were measured by an OC-EC aerosol analyzer (Sunset Laboratory Inc.). Water-soluble ions were analyzed by ion chromatography (Metrohm). All these species are major components in $PM_{2.5}$ (see Figure 2). Meteorological parameters (temperature and relative humidity) and gaseous pollutants ($SO_2$, CO, $NO_2$, NO, and $O_3$) at each site were recorded hourly. We further calculated the daily averages to probe the potential influence of air pollutants on BSOA formation.

For BSOA tracer analysis, detailed information of the processes is described in the previous literatures (Shen et al., 2015; Ding et al., 2012). Isotope-labeled standard mixtures, including dodecanoic acid-$d_{23}$, hexadecanoic acid-$d_{31}$, docosanoic acid-$d_{43}$ and levoglucosan-$^{13}C_6$ were added into each sample as internal standards. Then, samples were extracted by sonication with the mixed solvents of dichloride methane (DCM)/hexane (1:1, v/v) and DCM/methanol (1:1, v/v), sequentially. The extraction solutions

of each sample were combined, filtered, and concentrated to ~2 mL. Each concentrated sample was split
into two parts for silylation and methylation, respectively.
We analyzed fourteen BSOA tracers in the derivatized samples using GC/MSD (Agilent
7890/5975C). The isoprene-derived SOA ($SOA_I$) tracers were composed of 2-methyltetrols (2-MTLs, 2-
methylthreitol and 2-methylerythritol) (Claeys et al., 2004a), 2-methylglyceric acid (2-MGA) (Claeys et
al., 2004b), 3-MeTHF-3,4-diols (*cis*-3-methyltetrahydrofuran-3,4-diol and *trans*-3-
methyltetrahydrofuran-3,4-diol) (Lin et al., 2012) and $C_5$-alkene triols (*cis*-2-methyl-1,3,4-trihydroxy-1-
butene, *trans*-2-methyl-1,3,4-trihydroxy-1-butene and 3-methyl-2,3,4-trihydroxy-1-butene) (Wang et al.,
2005). The monoterpenes-derived SOA ($SOA_M$) tracers included 3-hydroxy-4,4-dimethylglutaric acid
(HDMGA), 3-hydroxyglutaric acid (HGA) (Claeys et al., 2007), pinic acid (PA), *cis*-pinonic acid (PNA)
(Christoffersen et al., 1998), and 3-methyl-1,2,3-butanetricarboxylic acid (MBTCA) (Szmigielski et al.,
2007). The β-caryophyllene-derived SOA ($SOA_C$) tracer was β-caryophyllenic acid (CA) (Jaoui et al.,
2007). Due to the lack of authentic standards, surrogate standards were used to quantify BSOA tracers
except PNA. Specifically, erythritol, PNA and octadecanoic acid were used for the quantification of $SOA_I$
tracers (Ding et al., 2008), other $SOA_M$ tracers (Ding et al., 2014) and CA (Ding et al., 2011), respectively.
The method detection limits (MDLs) for erythritol, PNA and octadecanoic acid were 0.01, 0.02, and 0.02
ng m$^{-3}$, respectively. Table S1 summarizes BSOA data at each site in the PRD.
**2.3 Quality Assurance / Quality Control**
These target BSOA tracers were not detected or lower than MDLs in the field blanks. The results of
spiked samples (erythritol, PNA and octadecanoic acid spiked in pre-baked quartz filters) indicated that
the recoveries were 65 ± 14 % for erythritol, 101 ± 3 % for PNA, and 83 ± 7 % for octadecanoic acid.
The results of paired duplicate samples indicated that all the relative differences for target BSOA tracers
were lower than 15%.
It should be noted that the application of surrogate quantification introduces additional errors to the
results. Based on the empirical approach to calculate uncertainties from surrogate quantification (Stone
et al., 2012), we estimated the errors in analyte measurement which were propagated from the
uncertainties in field blanks, spike recoveries, repeatability and surrogate quantification. As Table S2
showed, the estimated uncertainties in the tracers' measurement ranged from 15% (PNA) to 157% (CA).

## 3 Results and Discussion

### 3.1 PM$_{2.5}$ and gaseous pollutants

Figure 2 presents spatial and seasonal variations of PM$_{2.5}$ and its major components. Although annual-mean PM$_{2.5}$ (34.8 ± 6.1 μg m$^{-3}$) in the PRD met the NAAQS value of 35 μg m$^{-3}$, PM$_{2.5}$ at the urban sites (ZQ, GZ and DG) all exceeded the NAAQS value. The rural TH site in the northern part of PRD witnessed the lowest concentration of PM$_{2.5}$ (25.0 μg m$^{-3}$) among the nine sites. PM$_{2.5}$ levels were highest in winter (on average 60.1 ± 21.6 μg m$^{-3}$) and lowest in summer (on average 22.8 ± 3.3 μg m$^{-3}$). Carbonaceous aerosols and water-soluble ions together explained 98 ± 11 % of PM$_{2.5}$ masses. OM (OC×1.6) was the most abundant component in PM$_{2.5}$, followed by sulfate, ammonium, nitrate and EC. Similar to PM$_{2.5}$, the five major components all increased in winter and fall (Figure S1), suggesting severe PM$_{2.5}$ pollution during fall-winter season in the PRD.

In the gas phase, SO$_2$, CO, NO$_2$ and NO$_x$ presented similar seasonal trends as PM$_{2.5}$, i.e. higher levels occurred during fall and winter and lower concentrations during spring and summer (Figure 3 a-d). Annual-mean SO$_2$ and NO$_2$ in the PRD both met the NAAQS values of 60 μg m$^{-3}$ and 40 μg m$^{-3}$, respectively (Figure 3a and 3c). As a typical secondary pollutant, O$_3$ was highest in summer (Figure 3e), probably because of the strong photo-chemistry. Due to the compromise of opposite seasonal trends of O$_3$ and NO$_2$, O$_x$ showed less seasonal variation (Figure 3f) compared with other gaseous pollutants. And annual-mean O$_x$ reached 96.1 ± 14.9 μg m$^{-3}$. These indicated significant O$_3$ pollution all the year in the PRD.

### 3.2 Spatial distribution and seasonal variation of SOA tracers

The total concentrations of BSOA tracers ranged from 45.4 to 109 ng m$^{-3}$ among the nine sites. SOA$_M$ tracers (47.2 ± 9.29 ng m$^{-3}$) represented predominance, followed by SOA$_I$ tracers (23.1 ± 10.8 ng m$^{-3}$), and SOA$_C$ tracer (3.85 ± 1.75 ng m$^{-3}$).

### 3.2.1 Monoterpenes-derived SOA tracers

Annual averages of total SOA$_M$ tracers at the nine sites were in the range of 26.5 to 57.4 ng m$^{-3}$ (Table S1). Figure 4 and Figure S2a show the spatial distribution of SOA$_M$ tracers and monoterpene emissions in the PRD (Zheng et al., 2010). The highest concentration of SOA$_M$ tracers was observed at the rural TH site where monoterpene emissions were high. Figure 4 also presents seasonal variations of SOA$_M$

tracers. At most sites, high levels occurred in summer and fall. Monoterpene emission rates are
influenced by temperature and solar radiation (Guenther et al., 2012). Thus, high temperature and
intensive solar radiation during summer and fall in the PRD (Zheng et al., 2010) could stimulate
monoterpene emissions and then the $SOA_M$ formation.
Among the five $SOA_M$ tracers, HGA (20.1 ± 4.28 ng m$^{-3}$) showed the highest concentration,
followed by HDMGA (14.7 ± 2.93 ng m$^{-3}$), MBTCA (7.63 ± 1.49 ng m$^{-3}$), PNA (3.75 ± 2.72 ng m$^{-3}$) and
PA (1.01 ± 0.48 ng m$^{-3}$). $SOA_M$ formation undergoes multi-generation reactions. The first-generation
$SOA_M$ ($SOA_{M\_F}$) products, PNA and PA, can be further oxidized and form the later-generation ($SOA_{M\_L}$)
products, e.g. MBTCA (Müller et al., 2012). Thus, the (PNA+PA) / MBTCA ratio has been used to probe
$SOA_M$ aging (Haque et al., 2016; Ding et al., 2014). The (PNA+PA) / MBTCA ratios in chamber-
generated α-pinene SOA samples were reported in the range of 1.51 to 5.91 depending on different
oxidation conditions (Offenberg et al., 2007; Eddingsaas et al., 2012). In this study, the median values of
(PNA+PA) / MBTCA varied from 0.27 at ZH to 1.67 at TH. The ratios observed in this study were
consistent with our previous observations at the regional site, Wanqingsha (WQS) in the PRD (Ding et
al., 2012), but lower than those in the fresh α-pinene SOA samples from chamber experiments (Figure
S3), indicating relatively aged $SOA_M$ in the air of PRD.
Moreover, the levels of $SOA_{M\_L}$ tracers (HGA + HDMGA + MBTCA) were much higher than those
of $SOA_{M\_F}$ tracers (PNA + PA), with mean mass fractions of $SOA_{M\_L}$ tracers reaching 86% (Figure 4).
Mass fractions of $SOA_{M\_F}$ tracers decreased in the summer samples (Figure 4), probably resulting from
strong photo-chemistry and more intensive further oxidation during summer. High abundances of
$SOA_{M\_L}$ tracers in the PRD were different from our year-round observations at 12 sites across China
(Ding et al., 2016b). In that study, the (PNA+PA) / MBTCA ratio suggested generally fresh $SOA_M$ (Figure
S3) and $SOA_{M\_F}$ tracers were the majority. Thus, we see more aged $SOA_M$ in the PRD.
As Figure 5 a-b and Figure S4,S5 showed, the $SOA_{M\_F}$ tracers did not show good correlations with
$O_x$ at most sites, while the $SOA_{M\_L}$ tracers exhibited significant $O_x$ dependence. When $O_x$ is high, strong
photo-oxidation of PNA and PA could reduce their concentrations and promote the formation of $SOA_{M\_L}$
tracers (Müller et al., 2012). Thus, the levels of $SOA_{M\_L}$ tracers would increase with increasing $O_x$ but
not so for $SOA_{M\_F}$ tracers. On the other hand, sulfate is a key species in particles that determines aerosol
liquid water amount, aerosol acidity, and particle surface area (Xu et al., 2015, 2016). Thus, the increase
of sulfate could promote aqueous and heterogeneous reactions. In this study, the $SOA_{M\_F}$ tracers poorly
correlated with sulfate (Figure 5c), while the $SOA_{M\_L}$ tracers positively correlated with sulfate at all the
9 sites (Figure 5d). At each site the $SOA_{M\_L}$ tracers exhibited more sulfate dependence than the $SOA_{M\_F}$
tracers (Figure S5). This suggested that sulfate also played a critical role in forming $SOA_{M\_L}$ tracers
through the particle-phase reactions. Besides the gas-phase OH oxidation (Müller et al., 2012), the
heterogeneous OH oxidation of pinonic acid could also produce $SOA_{M\_L}$ tracers (Lai et al. 2015).
Aljawhary et al., (2016) reported the kinetics and mechanism of pinonic acid oxidation in acidic solutions
and found that the molar yields of MBTCA through the aqueous-phase reactions were similar to those in
the gas-phase oxidation. Here, we conclude that high concentrations of $O_x$ and sulfate could stimulate
$SOA_{M\_L}$ tracers' production and thereby lead to aged $SOA_M$ in the PRD.
**3.2.2 Isoprene-derived SOA tracers**
Annual averages of total $SOA_I$ tracers at the nine sites were in the range of 10.8 to 49.3 ng m$^{-3}$ (Table
S1). Figure 6 and Figure S2b show the spatial distribution of $SOA_I$ tracers and isoprene emissions in the
PRD (Zheng et al., 2010), respectively. The highest concentration occurred at ZQ where the emissions
were high. Figure 6 also presents seasonal variations of $SOA_I$ tracers at the nine sites. High levels
occurred in summer and fall. Similar to monoterpenes, the emission rate of isoprene is influenced by
temperature and solar radiation (Guenther et al., 2012), which are expected to be higher in summer and
fall in the PRD (Zheng et al., 2010). Among these $SOA_I$ tracers, 2-MTLs ($14.2 \pm 5.61$ ng m$^{-3}$) were the
most abundant products, followed by $C_5$-alkene triols ($6.81 \pm 5.05$ ng m$^{-3}$), 2-MGA ($1.99 \pm 0.72$ ng m$^{-3}$)
and 3-MeTHF-3,4-diols ($0.19 \pm 0.08$ ng m$^{-3}$).
$SOA_I$ formation is highly affected by $NO_x$ (Surratt et al., 2010). Under the low-$NO_x$ or $NO_x$ free
conditions, isoprene is oxidized by the OH and $HO_2$ radicals through the $HO_2$-channel which generates
a hydroxy hydroperoxide (ISOPOOH) and then forms epoxydiols (IEPOX) (Paulot et al., 2009). Reactive
uptake of IEPOX on acidic particles eventually produces 2-MTLs, $C_5$-alkene triols, 3-MeTHF-3,4-diols,
2-MTLs-organosulfates and oligomers (Lin et al., 2012). Under the high-$NO_x$ conditions, isoprene
undergoes oxidation by $NO_x$ through the NO/$NO_2$-channel and generates methacrolein (MACR) and then
forms peroxymethylacrylic nitric anhydride (MPAN). Further oxidation of MPAN by the OH radical
produces hydroxymethel-methyl-α-lactone (HMML) and/or methacrylic acid epoxide (MAE). HMML
and MAE are the direct precursors to 2-MGA, 2-MGA-organosulfate and its corresponding oligomers
(Nguyen et al., 2015). As Figure 6 showed, the concentrations of $HO_2$-channel tracers (2-MTLs + $C_5$-
alkene triols + 3-MeTHF-3,4-diols) were much higher than those of the NO/NO$_2$-channel product (2-
MGA) at all the nine sites. The dominance of HO$_2$-channel products was also observed at another
regional site in the PRD (WQS) (He et al., 2018).
Figure 6 also shows seasonal trends of the 2-MGA to 2-MTLs ratio (2-MGA/2-MTLs) which is
often applied to probe the influence of NO$_x$ on the formation of SOA$_I$ (Ding et al., 2013; Ding et al.,
2016a; Pye et al., 2013). The ratios were highest in wintertime and lowest in summertime, which were
consistent with the seasonal trend of NO$_x$ during our campaign (Figure 3d). As Table 1 showed, 2-MGA
positively correlated with NO$_2$, probably due to the enhanced formation of MPAN from
peroxymethacryoyl (PMA) radical reacted with NO$_2$ (Worton et al., 2013; Chan et al., 2010). Previous
laboratory studies showed that increasing NO$_2$/NO ratio could promote the formation of 2-MGA and its
corresponding oligoesters (Chan et al., 2010; Surratt et al., 2010). However, we did not see a significant
correlation between 2-MGA and NO$_2$/NO ratio in the PRD. Instead, the 2-MGA/2-MTLs ratio correlated
well with NO, NO$_2$ and NO$_2$/NO ratio (Table 1). Increasing NO limits the formation of ISOPOOH but
prefers the production of MACR, and increasing NO$_2$ enhances MPAN formation. Thus, it is expected
that the 2-MGA/2-MTLs ratio shows stronger NO$_x$ dependence than 2-MGA. These findings demonstrate
the significant impact of NO$_x$ on SOA$_I$ formation pathways in the atmosphere. We also checked the
correlations of SOA$_I$ tracers with O$_x$ and sulfate (Figure S6). The NO/NO$_2$-channel product exhibited
more O$_x$ and sulfate dependance than HO$_2$-channel products.
Recent studies indicated that isoprene ozonolysis might play a role in SOA$_I$ formation in the ambient
air. Riva et al. (2016) found that isoprene ozonolysis with acidic particles could produce substantial 2-
MTLs but not so for C$_5$-alkene triols and 3-MeTHF-3,4-diols. Li et al. (2018) observed a positive
correlation between 2-MTLs and O$_3$ in the North China Plain. In the PRD, we also saw weak but
significant correlations of 2-MTLs with O$_3$ (Table S3). However, 3-MeTHF-3,4-diols and C$_5$-alkene
triols were detected in all samples and 2-MTLs, C$_5$-alkene triols and 3-MeTHF-3,4-diols correlated well
with each other (Table S4), which was apparently different from those reported by Riva et al. (2016).
Moreover, the ratios of 2-MTLs isomers in the PRD samples (2.00−2.85) were much lower than those
(10−22, Figure S7) reported in the SOA from isoprene ozonolysis (Riva et al., 2016). Furthermore,
isoprene oxidation by the OH radical is much faster than that by ozone under the polluted PRD conditions
(Table S5). And IEPOX yields through the ISOPOOH oxidation by the OH radical are more than 75%
in the atmosphere (St. Clair et al., 2016). Thus, isoprene ozonolysis might be not the major formation
pathway of $SOA_I$, even though annual-mean $O_3$ level reaching 67.7 µg m$^{-3}$ in the PRD (Table S1).

Previous studies found that thermal decomposition of low volatility organics in IEPOX-derived

SOA could produce $SOA_I$ tracers, e.g. 2-MTLs, $C_5$-alkene triols and 3-MeTHF-3,4-diols (Lopez-Hilfiker
et al., 2016, Watanabe et al., 2018). This means that these tracers detected by GC-MSD might be
generated from thermal decomposition of IEPOX-derived SOA. As estimated by Cui et al (2018), 14.7-
42.8% of $C_5$-alkene triols, 11.1% of 2-MTLs and approximately all 3-MeTHF-3,4-diols.measured by
GC/ MSD could be attributed to the thermal degradation of 2-MTLs-derived organosulfates (MTL-OSs).
We also measured MTL-OSs in two samples at HS and TS sites, respectively (Table S6) using the widely
used LC-MS approach (He et al., 2014, 2018). Assuming that all MTL-OSs decomposed to these tracers,
the thermal decomposition of MTL-OSs would account for 15.1-31.6% of $C_5$-alkene triols, 6.0-10.0% of
2-MTLs and all 3-MeTHF-3,4-diols measured by GC/ MSD. Thus, $C_5$-alkene triols and 2-MTLs are
major from isoprene oxidation rather than thermal decomposition of MTL-OSs, while 3-MeTHF-3,4-
diols are only in trace amount in the air and might be produced largely from thermal degradation.

Moreover, we see significant variations in $SOA_I$ tracer compositions in the PRD. For instant, $C_5$-

alkene triols have three isomers. If these tracers were mainly generated from a thermal process, their
compositions should be similar in different samples. In fact, the relative abundances of three $C_5$-alkene
triol isomers significantly changed from site to site (Figure 7) and season to season (Figure S8), and their
compositions in the PRD were different from those measured in the chamber samples (Lin et al., 2012).
In addition, the slopes of linear correlations among these IEPOX-derived SOA tracers also varied from
site to site (Figure S9). Coupled with the seasonal trend of 2-MGA/2-MTLs ratios, the apparent variations
in $SOA_I$ tracer compositions demonstrate that these $SOA_I$ tracers are mainly formed through different
pathways in the ambient atmosphere, although part of them might arise from the thermal decomposition
of different dimers/OSs and the parent dimers/OSs varies with sites and seasons.
**3.2.3 Sesquiterpene-derived SOA tracer**
Annual averages of CA at the nine sites ranged from 1.82 to 7.07 ng m$^{-3}$. The levels of CA at the inland
sites (e.g. GZ, ZQ, and TH) were higher than those at the coastal sites (ZH and NS, Figure 8). Since
sesquiterpenes are typical BVOCs, it is unexpected that the concentrations of CA were highest during
winter in the PRD (Figure 8). Interestingly, seasonal trend of CA was consistent with that of the biomass
burning (BB) tracer, levoglucosan (Figure 8). And CA correlated well with levoglucosan at eight sites in
the PRD (Figure 9a). Sesquiterpenes are stored in plant tissues partly to protect the plants from insects
and pathogens (Keeling and Bohlmann, 2006). BB can not only stimulate sesquiterpene emissions
(Ciccioli et al., 2014) but also substantially alter the SOA formation and yields (Mentel et al., 2013).
Emissions inventories in the PRD showed that the BB emissions were enhanced during winter (He et al.,
2011). These suggested that the unexpected increase of $SOA_C$ in wintertime could be highly associated
with BB emissions in the PRD.

Besides the impact of BB, we also found positive correlations of CA with $O_x$ (Figure 9b) and sulfate

(Figure 9c). The oxidation of β-caryophyllene by the OH radical and $O_3$ is very rapid. Under typical
oxidation conditions in the air of PRD, the lifetimes of β-caryophyllene are only several minutes (Table
S5). Once emitted from vegetation or biomass burning, β-caryophyllene will reacted rapidly and form
CA immediately. This partly explains the positive correlations between CA and levoglucosan in the PRD.
The unexpected high levels of CA in the winter indicated that biomass burning could be an important
source of $SOA_C$ in the PRD, especially in wintertime. In addition, the increase of sulfate could raise
aerosol acidity and thereby promote aqueous and heterogeneous reactions to form $SOA_C$. In the PRD,
both $O_x$ (Figure 3f) and sulfate (Figure S1) increased during winter, which could promote $SOA_C$
formation. Here, we conclude that the enhancement of BB emissions as well as the increase of $O_x$ and
sulfate in wintertime together led to high $SOA_C$ production during winter in the PRD.
**3.3 Source apportionment and atmospheric implications**
We further attributed BSOA by the SOA-tracer approach which was first developed by Kleindienst et al.
(2007). This method has applied to SOA apportionment at multiple sites across the United States
(Lewandowski et al., 2013) and China (Ding et al., 2016b), and over global oceans from Arctic to
Antarctic (Hu et al., 2013). Details of the SOA-tracer method and its application in this study as well as
the uncertainty of estimating procedure are described in Text S1. Table S1 lists the results of estimated
SOA from different BVOCs.

Figure 10a exhibits the spatial distribution of BSOA ($SOA_M$ + $SOA_I$ + $SOA_C$). Annual average at

the nine sites ranged from 0.97 μg m$^{-3}$ (NS) to 2.19 μg m$^{-3}$ (ZQ), accounting for 9-15% of OM. $SOA_M$
was the largest BSOA contributor with an average contribution of 64 ± 7 %, followed by $SOA_C$ (21 ±
6 %), and $SOA_I$ (14 ± 4 %). Figure 10b presents seasonal variation of BSOA. The levels were highest in
fall (2.35 ± 0.95 μg m$^{-3}$) and lowest in spring (1.06 ± 0.42 μg m$^{-3}$). $SOA_M$ contributions ranged from 57%

307 in winter to 68% in spring. The shares of $SOA_I$ were only 5% in winter and reached up to 22% in summer.

308 The contributions of $SOA_C$ increased to 40% in wintertime.

309  It is interesting to note that $SOA_M$, $SOA_I$ and $SOA_C$ all positively correlated with sulfate and $O_x$ in

310 the PRD (Table 2). Since anthropogenic emissions can enhance BSOA formation (Hoyle et al., 2011),

311 the reduction of anthropogenic emissions indeed lowers BSOA production (Carlton et al., 2018). As the

312 oxidation product of $SO_2$, sulfate is a key species in particles that determines aerosol acidity and surface

313 areas (Xu et al., 2015, 2016) which could promotes BSOA formation through the acid-catalyzed

314 heterogeneous reactions. Recent study found that $SO_2$ could directly reaction with organic peroxides of

315 monoterpene ozonolysis and form substantial organosulfates (Ye et al., 2018). Thus, the decrease of $SO_2$

316 emission indeed reduces $SO_2$ and sulfate in the ambient air, which hereby leads to less acidic particles

317 and reduces the BSOA production. For $O_x$, the increase of $O_3$ likely results in significant SOA formation

318 through the BVOCs ozonolysis (Sipilä et al., 2014; Riva et al., 2017). Hence, the decrease of $O_x$ resulting

319 from the control of VOCs and $NO_x$ emissions could reduce BSOA formation through $O_3$ chemistry. Based

320 on the observed sulfate and $O_x$ dependence of BSOA in this study, the reduction of 1 μg m$^{-3}$ in sulfate

321 and $O_x$ in the air of PRD could lower BSOA levels by 0.17 and 0.02 μg m$^{-3}$, respectively. If both

322 concentrations decline by 50%, the reduction of $O_x$ is more efficient than sulfate in reducing BSOA in

323 the PRD (Table 2).

324  We further compared the results in 2015 with those during fall-winter season in 2008 at WQS (Ding

325 et al., 2012). We found that all BSOA species positively correlated with sulfate but exhibited no $O_x$

326 dependence (Table S7). Thus, in 2008 BSOA formation was largely influenced by sulfate, probably due

327 to high sulfate levels then (as high as 46.8 μg m$^{-3}$). Owing to strict control of $SO_2$ emissions (Wang et al.,

328 2013), ambient $SO_2$ significantly shrank over the PRD (Figure 1b). Our long-term observation during

329 fall-winter season at WQS also witnessed a decreasing trend of sulfate from 2007 to 2016 (Figure S10).

330 However, $O_x$ levels did not decrease during the past decade (Figure 1b) and $O_x$ concentrations were much

331 higher than sulfate in 2015 in the PRD (96.1 ± 14.9 μg m$^{-3}$ vs. 8.44 ± 1.09 μg m$^{-3}$ on average). All these

332 underline the importance of $O_x$ in BSOA formation currently in the PRD. At present, short-term despiking

333 and long-term attainment of $O_3$ concentrations are challenges for air pollution control in the PRD (Ou et

334 al., 2016). Thus, lowering $O_x$ is critical to improve air quality in the PRD. Our results highlight the

335 importance of future reduction in anthropogenic pollutant emissions (e.g. $SO_2$ and $O_x$ precursors) for

336 considerably reducing the BSOA burden in polluted regions.

**Code/Data availability**

The experimental data in this study are available upon request to the corresponding author by email.

**Author Contribution**

Xiang Ding, Duo-Hong Chen and Jun Li conceived the project and designed the study. Yu-Qing Zhang and Duo-Hong Chen performed the data analysis and wrote the manuscript. Duo-Hong Chen, Tao Zhang and Yu-Bo Ou arranged the sample collection and assisted with the data analysis. Jun-Qi Wang, Qian Cheng and Hao Jiang analyzed the samples. Xiang Ding, Peng-Lin Ye, Wei Song, Gan Zhang and Xin-Ming Wang performed data interpretation and edited the manuscript. All authors contributed to the final manuscript development.

**Competing interests**

The authors declare that they have no conflict of interest.

**Acknowledgments**

This study was supported by National Key Research and Development Program (2018YFC0213902), National Natural Science Foundation of China (41722305/41603070/41473099), and Local Innovative and Research Teams Project of Guangdong Pearl River Talents Program (NO. 2017BT01Z134). We would like to thank Prof. Sasho Gligorovski for his helpful suggestion on the discussion of atmospheric oxidation process. The data of gaseous pollutants, major components in $PM_{2.5}$ and BSOA tracers can be found in supporting information.

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

Contributions of toluene and α-pinene to SOA formed in an irradiated toluene/α-pinene/$NO_x$/air
mixture: Comparison of results using [14]C content and SOA organic tracer methods, Environ. Sci.
Technol., 41, 3972-3976, 10.1021/es070089+, 2007.
Offenberg, J. H., Lewandowski, M., Edney, E. O., Kleindienst, T. E., and Jaoui, M.: Influence of aerosol
acidity on the formation of secondary organic aerosol from biogenic precursor hydrocarbons, Environ.
Sci. Technol., 43, 7742–7747, 10.1021/es901538e, 2009.
Ou, J., Yuan, Z., Zheng, J., Huang, Z., Shao, M., Li, Z., Huang, X., Guo, H., and Louie, P. K. K.: Ambient
ozone control in a photochemically active region: Short-term despiking or long-term attainment?,
Environ. Sci. Technol., 50, 5720–5728, 10.1021/acs.est.6b00345, 2016.
Paulot, F., Crounse, J. D., Kjaergaard, H. G., Kürten, A., St. Clair, J. M., Seinfeld, J. H., and Wennberg,
P. O.: Unexpected epoxide formation in the gas-phase photooxidation of isoprene, Science, 325, 730-
733, 10.1126/science.1172910, 2009.
Pye, H. O. T., Pinder, R. W., Piletic, I. R., Xie, Y., Capps, S. L., Lin, Y.-H., Surratt, J. D., Zhang, Z., Gold,
A., Luecken, D. J., Hutzell, W. T., Jaoui, M., Offenberg, J. H., Kleindienst, T. E., Lewandowski, M.,
and Edney, E. O.: Epoxide pathways improve model predictions of isoprene markers and reveal key
role of acidity in aerosol formation, Environ. Sci. Technol., 47, 11056-11064, 10.1021/es402106h,

2013.

Riva, M., Budisulistiorini, S. H., Zhang, Z., Gold, A., and Surratt, J. D.: Chemical characterization of
secondary organic aerosol constituents from isoprene ozonolysis in the presence of acidic aerosol,
Atmos. Environ., 130, 5-13, 10.1016/j.atmosenv.2015.06.027, 2016.
Riva, M., Budisulistiorini, S. H., Zhang, Z., Gold, A., Thornton, J. A., Turpin, B. J., and Surratt, J. D.:
Multiphase reactivity of gaseous hydroperoxide oligomers produced from isoprene ozonolysis in the
presence of acidified aerosols, Atmos. Environ., 152, 314-322, 10.1016/j.atmosenv.2016.12.040, 2017.
Shen, R. Q., Ding, X., He, Q. F., Cong, Z. Y., Yu, Q. Q., and Wang, X. M.: Seasonal variation of secondary
organic aerosol tracers in Central Tibetan Plateau, Atmos. Chem. Phys., 15, 8781-8793, 10.5194/acp-

15-8781-2015, 2015.

Sipilä, M., Jokinen, T., Berndt, T., Richters, S., Makkonen, R., Donahue, N. M., Mauldin Iii, R. L., Kurtén,
T., Paasonen, P., Sarnela, N., Ehn, M., Junninen, H., Rissanen, M. P., Thornton, J., Stratmann, F.,
Herrmann, H., Worsnop, D. R., Kulmala, M., Kerminen, V. M., and Petäjä, T.: Reactivity of stabilized
Criegee intermediates (sCIs) from isoprene and monoterpene ozonolysis toward $SO_2$ and organic acids,
Atmos. Chem. Phys., 14, 12143-12153, 10.5194/acp-14-12143-2014, 2014.
St. Clair, J. M., Rivera-Rios, J. C., Crounse, J. D., Knap, H. C., Bates, K. H., Teng, A. P., Jørgensen, S.,
Kjaergaard, H. G., Keutsch, F. N., and Wennberg, P. O.: Kinetics and products of the reaction of the
first-generation isoprene hydroxy hydroperoxide (ISOPOOH) with OH, J. Phys. Chem. A., 120, 1441-
1451, 10.1021/acs.jpca.5b06532, 2016.
Stone, E. A., Nguyen, T. T., Pradhan, B. B., and Man Dangol, P.: Assessment of biogenic secondary
organic aerosol in the Himalayas, Environ. Chem., 9, 263-272, 10.1071/EN12002, 2012.
Surratt, J. D., Chan, A. W. H., Eddingsaas, N. C., Chan, M., Loza, C. L., Kwan, A. J., Hersey, S. P.,
Flagan, R. C., Wennberg, P. O., and Seinfeld, J. H.: Reactive intermediates revealed in secondary
organic aerosol formation from isoprene, P. Natl. Acad. Sci. USA., 107, 6640-6645,
10.1073/P.Natl.Acad.Sci.USA.0911114107, 2010.
Szmigielski, R., Surratt, J. D., Gómez-González, Y., Van der Veken, P., Kourtchev, I., Vermeylen, R.,
Blockhuys, F., Jaoui, M., Kleindienst, T. E., Lewandowski, M., Offenberg, J. H., Edney, E. O., Seinfeld,
J. H., Maenhaut, W., and Claeys, M.: 3-methyl-1,2,3-butanetricarboxylic acid: An atmospheric tracer
for terpene secondary organic aerosol, Geophys. Res. Let., 34, 10.1029/2007gl031338, 2007.
Von Schneidemesser, E., Monks, P. S., Allan, J. D., Bruhwiler, L., Forster, P., Fowler, D., Lauer, A.,
Morgan, W. T., Paasonen, P., Righi, M., Sindelarova, K., and Sutton, M. A.: Chemistry and the linkages
between air quality and climate change, Chem. Rev., 115, 3856-3897, 10.1021/acs.chemrev.5b00089,

2015.

Wang, W., Kourtchev, I., Graham, B., Cafmeyer, J., Maenhaut, i., and Claeys, M.: Characterization of

oxygenated derivatives of isoprene related to 2-methyltetrols in Amazonian aerosols using

trimethylsilylation and gas chromatography/ion trap mass spectrometry, Rapid Commun. Mass

Spectrom., 19, 1343-1351, 10.1002/rcm.1940, 2005.

Wang, X., Liu, H., Pang, J., Carmichael, G., He, K., Fan, Q., Zhong, L., Wu, Z., and Zhang, J.: Reductions

in sulfur pollution in the Pearl River Delta region, China: Assessing the effectiveness of emission

controls, Atmos. Environ., 76, 113-124, 10.1016/j.atmosenv.2013.04.074, 2013.

Watanabe, A. C., Stropoli, S. J., and Elrod, M. J.: Assessing the potential mechanisms of isomerization

reactions of isoprene epoxydiols on secondary organic aerosol, Environ. Sci. Technol., 52, 8346-8354,

10.1021/acs.est.8b01780, 2018.

Worton, D. R., Surratt, J. D., LaFranchi, B. W., Chan, A. W. H., Zhao, Y., Weber, R. J., Park, J.-H.,

Gilman, J. B., de Gouw, J., Park, C., Schade, G., Beaver, M., Clair, J. M. S., Crounse, J., Wennberg,

P., Wolfe, G. M., Harrold, S., Thornton, J. A., Farmer, D. K., Docherty, K. S., Cubison, M. J., Jimenez,

556        J.-L., Frossard, A. A., Russell, L. M., Kristensen, K., Glasius, M., Mao, J., Ren, X., Brune, W., Browne,

E. C., Pusede, S. E., Cohen, R. C., Seinfeld, J. H., and Goldstein, A. H.: Observational insights into

aerosol formation from isoprene, Environ. Sci. Technol., 47, 11396–11402, 10.1021/es4011064, 2013.

Xu, L., Guo, H., Boyd, C. M., Klein, M., Bougiatioti, A., Cerully, K. M., Hite, J. R., Isaacman-VanWertz,

G., Kreisberg, N. M., Knote, C., Olson, K., Koss, A., Goldstein, A. H., Hering, S. V., de Gouw, J.,

Baumann, K., Lee, S.-H., Nenes, A., Weber, R. J., and Ng, N. L.: Effects of anthropogenic emissions

on aerosol formation from isoprene and monoterpenes in the southeastern United States, P. Natl. Acad.

Sci. USA., 112, 37-42, 10.1073/ P.Natl.Acad.Sci.USA.1417609112, 2015.

Xu, L., Middlebrook, A. M., Liao, J., de Gouw, J. A., Guo, H., Weber, R. J., Nenes, A., Lopez-Hilfiker,

F. D., Lee, B. H., Thornton, J. A., Brock, C. A., Neuman, J. A., Nowak, J. B., Pollack, I. B., Welti, A.,

Graus, M., Warneke, C., and Ng, N. L.: Enhanced formation of isoprene-derived organic aerosol in

sulfur-rich power plant plumes during Southeast Nexus, J. Geophys. Res-Atmos., 121, 11137-11153,

10.1002/2016JD025156, 2016.

Ye, J., Abbatt, J. P. D., and Chan, A. W. H.: Novel pathway of $SO_2$ oxidation in the atmosphere: Reactions

with monoterpene ozonolysis intermediates and secondary organic aerosol, Atmos. Chem. Phys., 18,

5549-5565, 10.5194/acp-18-5549-2018, 2018.

Zhang, R. Y., Wang, G. H., Guo, S., Zarnora, M. L., Ying, Q., Lin, Y., Wang, W. G., Hu, M., and Wang,
Y.: Formation of urban fine particulate matter, Chem. Rev., 115, 3803-3855,
10.1021/acs.chemrev.5b00067, 2015.
Zheng, J., Zheng, Z., Yu, Y., and Zhong, L.: Temporal, spatial characteristics and uncertainty of biogenic
VOC emissions in the Pearl River Delta region, China, Atmos. Environ., 44, 1960-1969,
10.1016/j.atmosenv.2010.03.001, 2010.

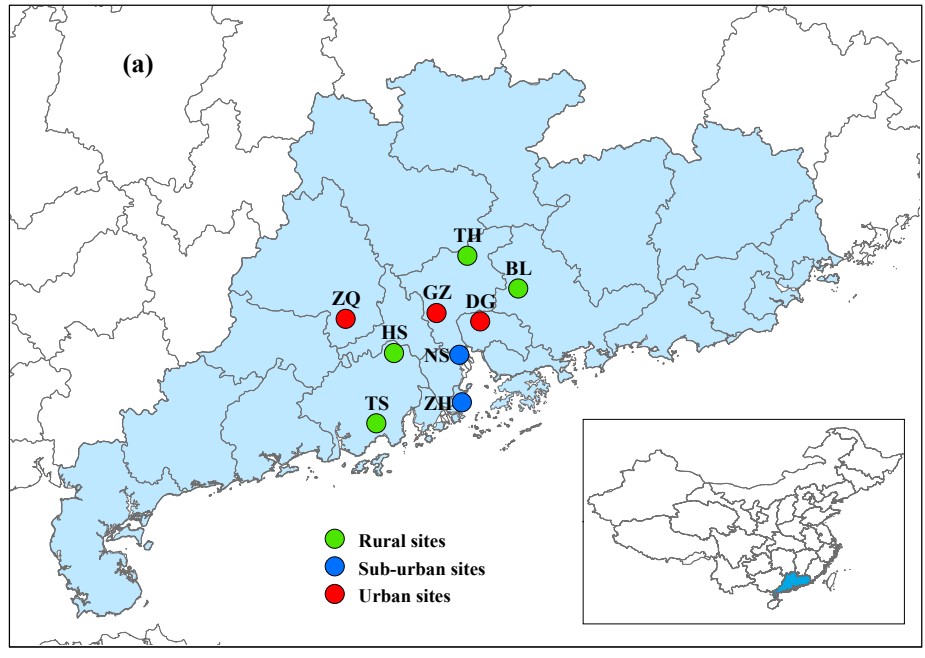


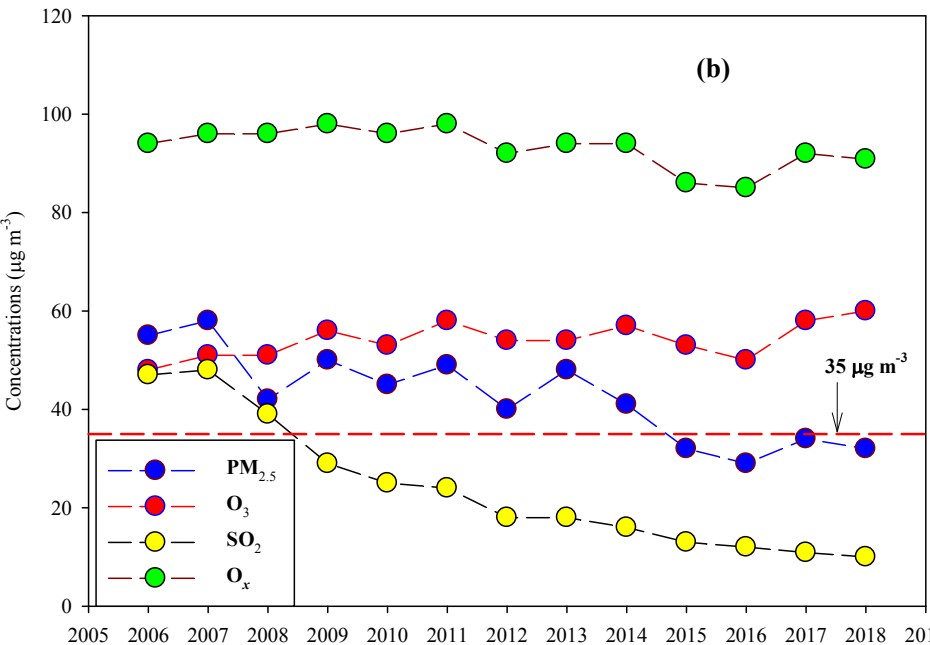


Figure 1 Sampling sites in the PRD (a) and long-term trends of annual-mean $PM_{2.5}$, $O_3$, $SO_2$ and $O_x$ recorded by the

Guangdong-Hong Kong-Macao regional air quality monitoring network (http://www.gdep.gov.cn/hjjce/) (b). The

red dash line indicates the NAAQS for annual-mean $PM_{2.5}$ concentrations (35 μg m⁻³).


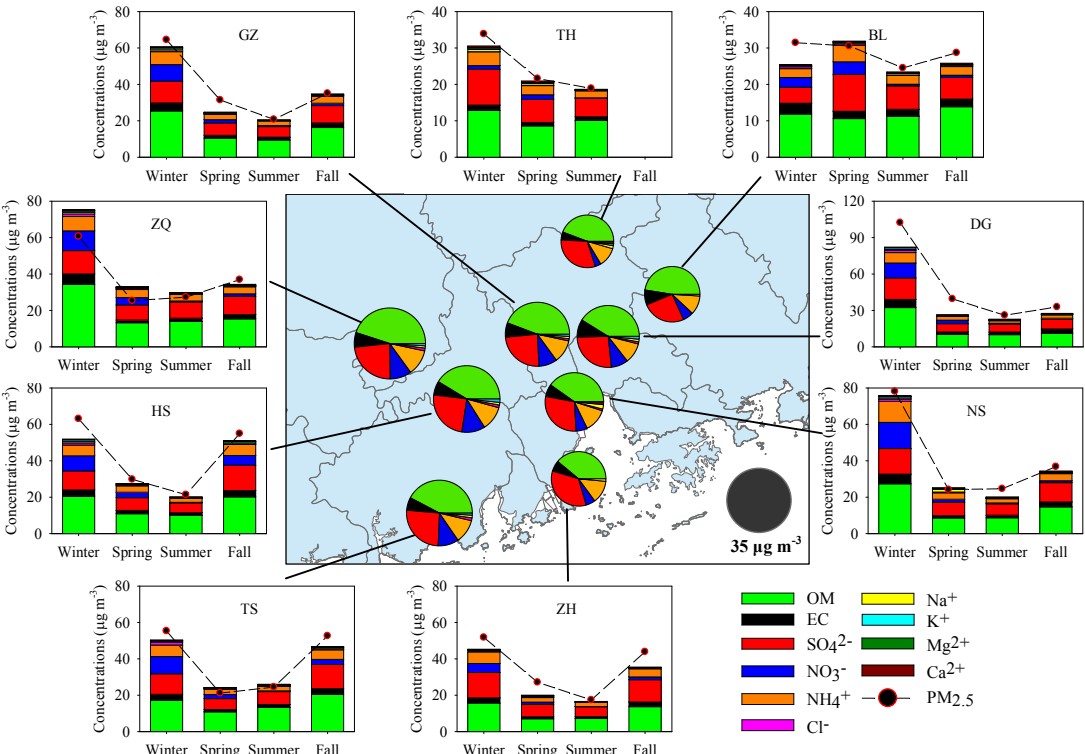

Figure 2 Major components in PM$_{2.5}$ and their seasonal variation at 9 sites. The pie charts in the central figure represent the annual average of major components. High levels of PM$_{2.5}$ and major components were observed in wintertime.

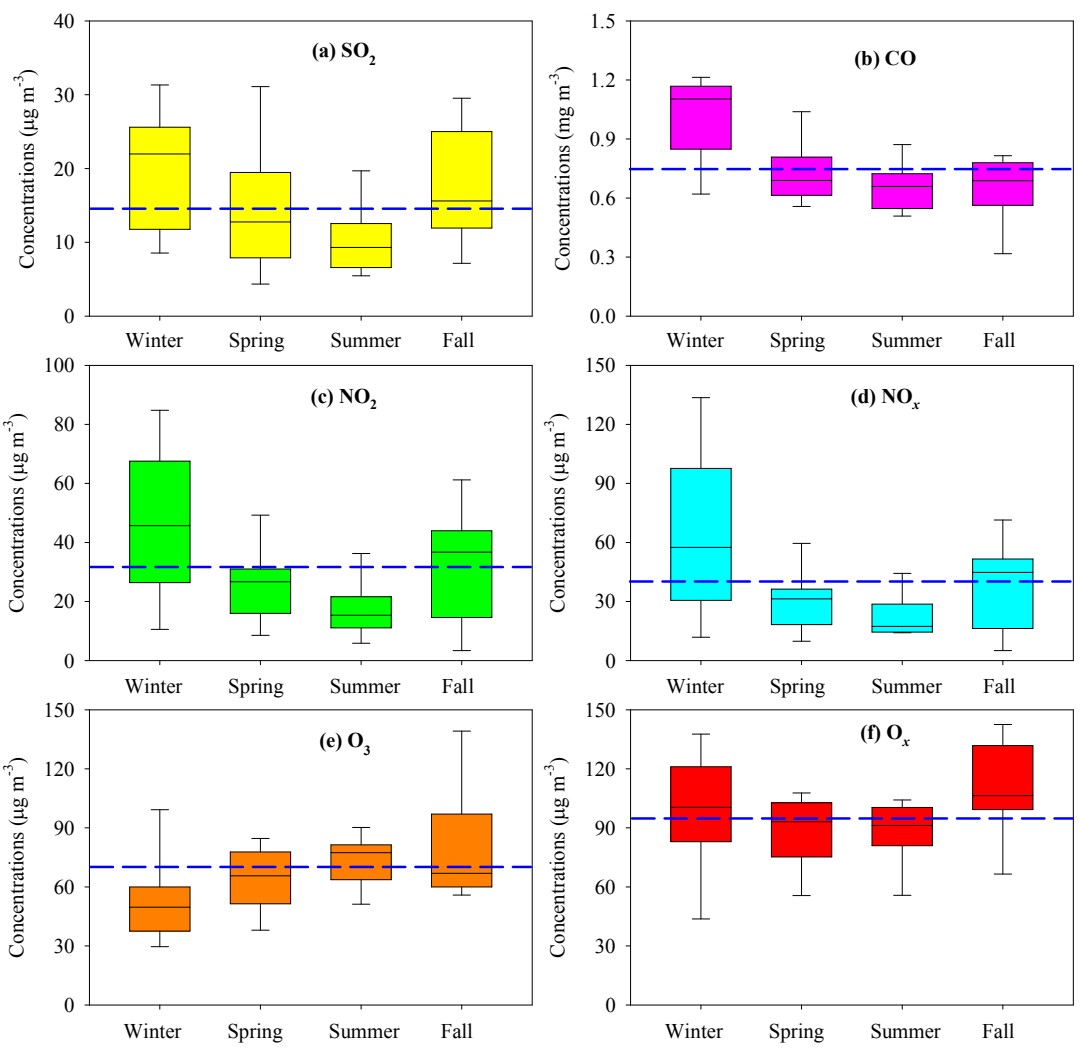


Figure 3 Seasonal variation of gaseous pollutants in the PRD. Box with error bars represent 10th, 25th, 75th, 90th
percentiles for each pollutant. The line in each box represents the median value. Blue dash lines indicate annual
average concentrations of $SO_2$ (14.9 μg m⁻³), CO (0.74 mg m⁻³), $NO_2$ (28.5 μg m⁻³), $NO_x$ (39.0μg m⁻³), $O_3$ (67.7 μg
m⁻³) and $O_x$ (96.1 μg m⁻³).

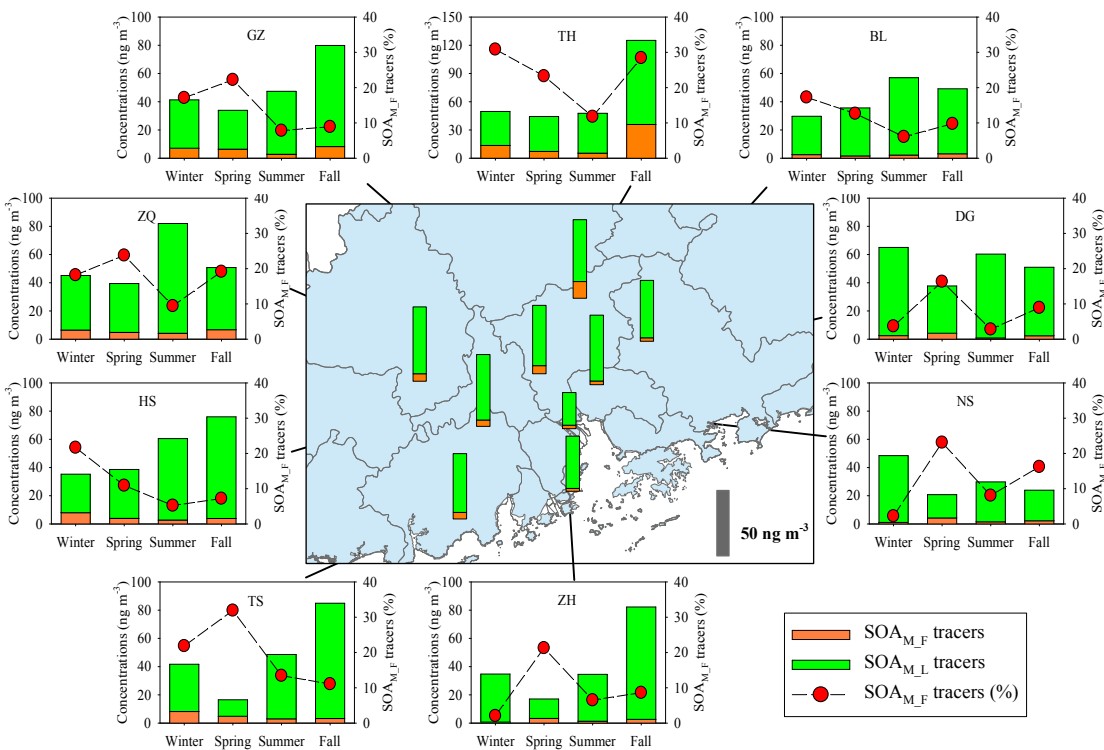


Figure 4 Spatial and seasonal variations of SOA$_M$ tracers at 9 sites in the PRD. The bars in the central figure represent

the annual average concentrations of the SOA$_M$ tracers.


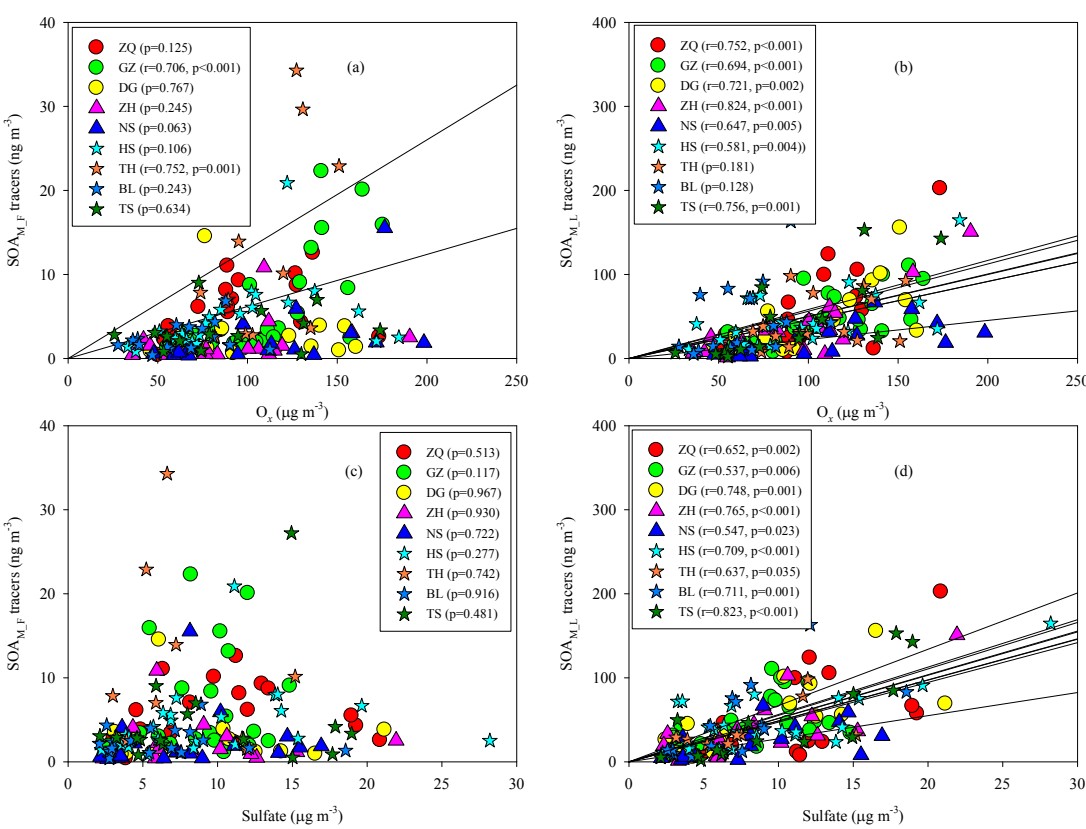


Figure 5 Correlations of SOA$_{M\_F}$ and SOA$_{M\_L}$ tracers with O$_x$ (a, b) and sulfate (c, d)

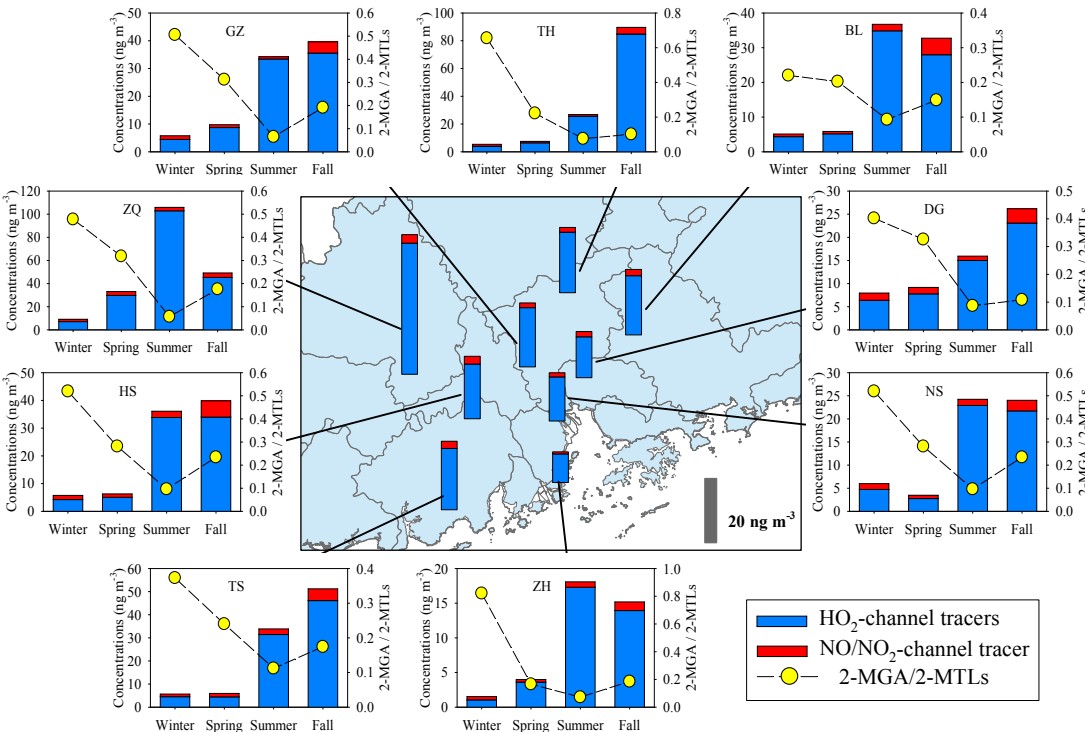

Figure 6 Spatial and seasonal variations of SOA$_I$ tracers at 9 sites in the PRD. The bars in the central figure represent
the annual average concentrations of the SOA$_I$ tracers.

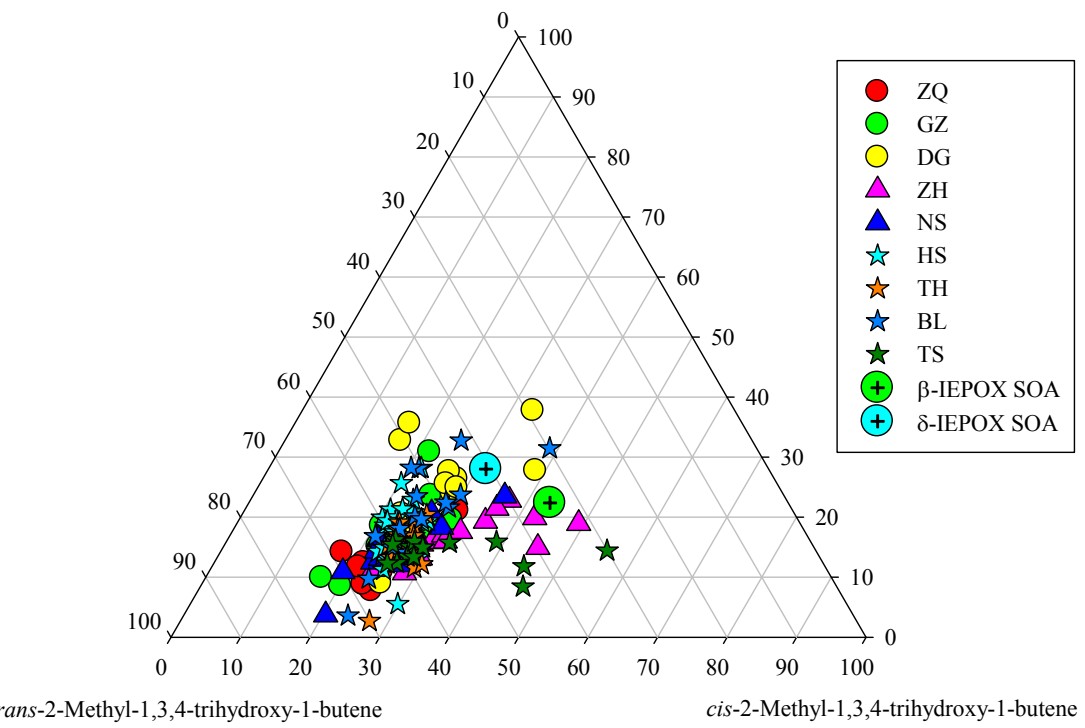


Figure 7 Ternary plot of $C_5$-alkene triol isomers in the PRD samples and in the β-IEPOX and δ-IEPOX derived

SOA (Lin et al., 2012).


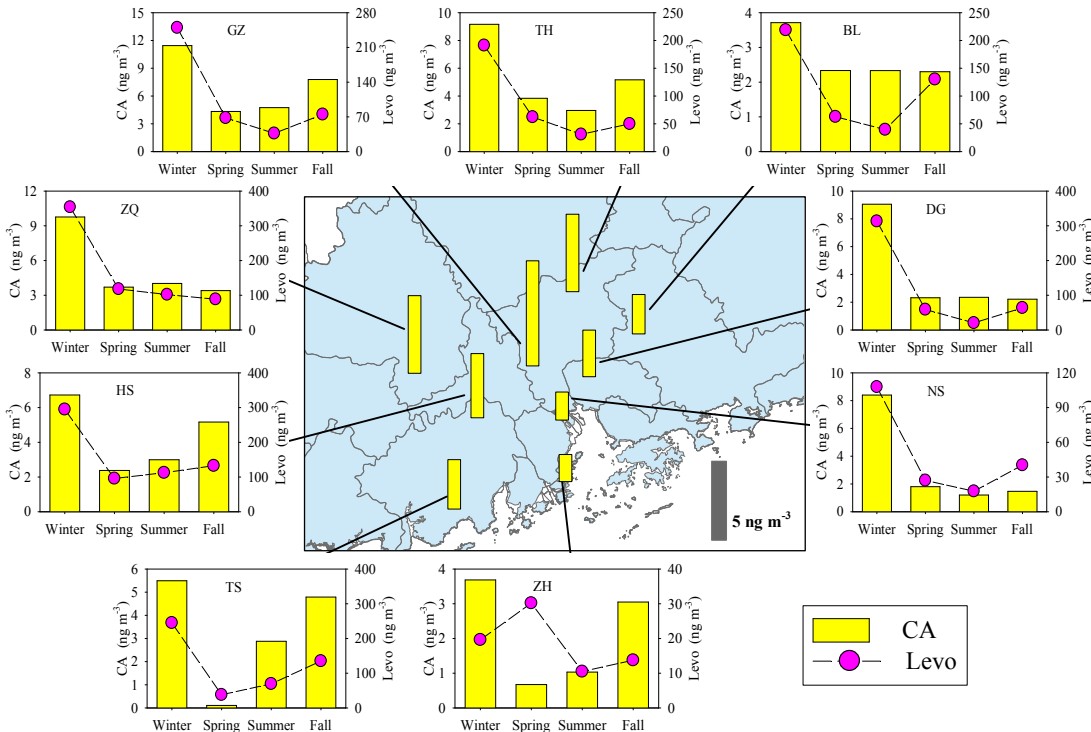

Figure 8 Spatial and seasonal variations of SOA$_c$ tracer (CA) at 9 sites in the PRD. The bars in the central figure
represent the annual average concentration of the SOA$_c$ tracers. The pink circle indicates the BB tracer, levoglucosan
(Levo).

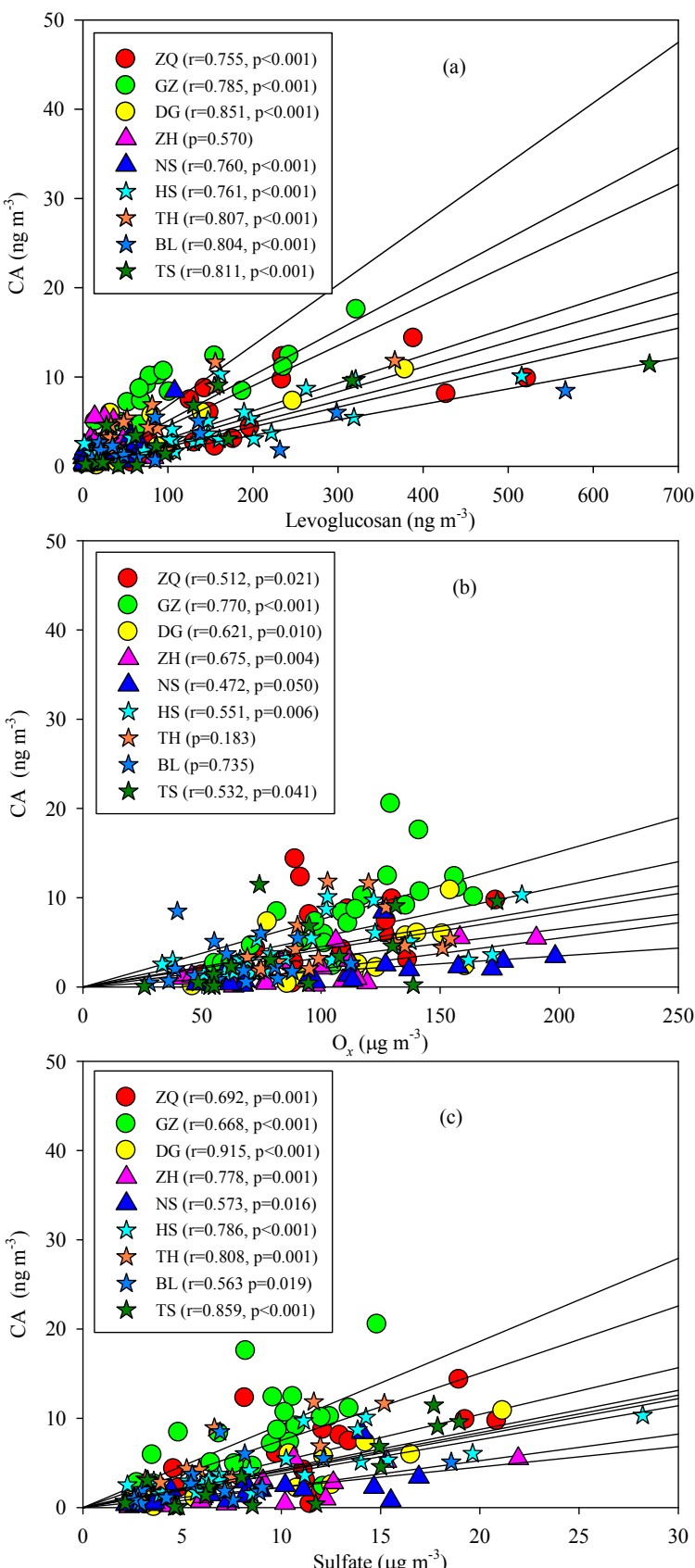


Figure 9 Significant correlations of CA with levoglucosan (a), $O_x$ (b) and sulfate (c).


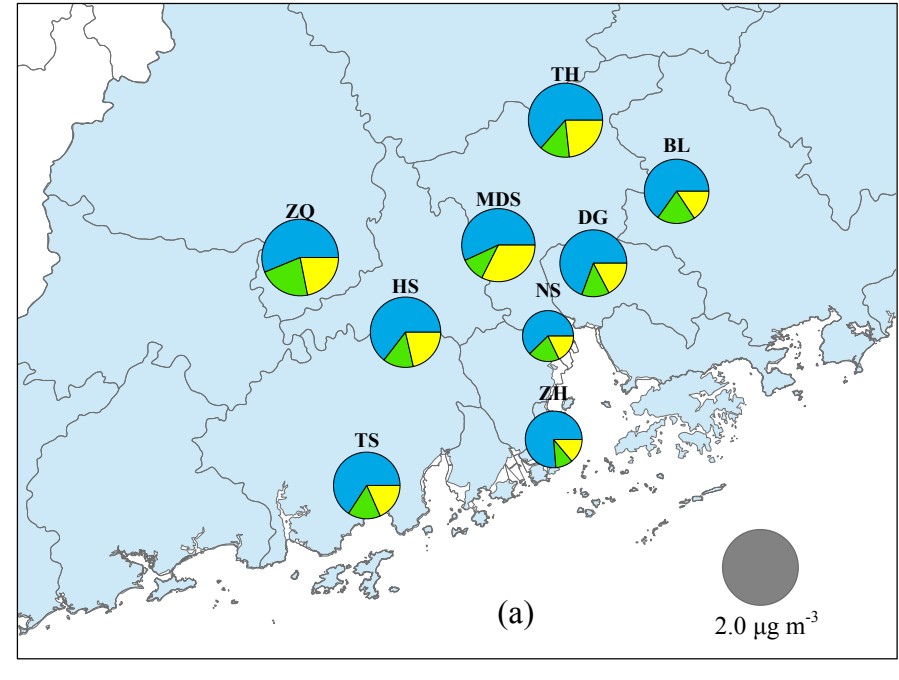


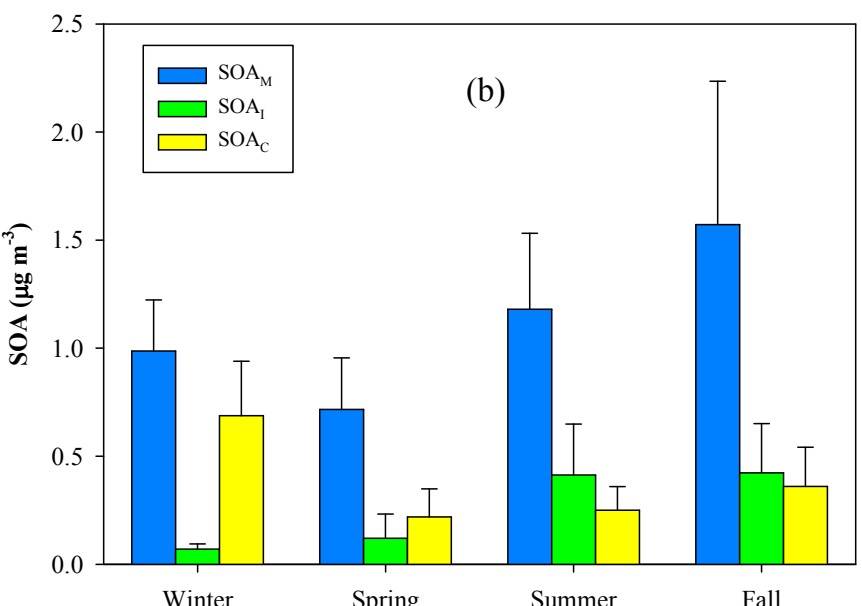

Figure 10 Spatial (a) and seasonal (b) variations in BSOA components

Table 1 Correlations between $SOA_I$ tracers and $NO_x$

| | 2-MGA | | 2-MGA/2-MTLs | |
|---|---|---|---|---|
| | Coefficient (r) | $p$-value | Coefficient (r) | $p$-value |
| NO | 0.028 | 0.733 | 0.166 | 0.043 |
| $NO_2$ | 0.205 | 0.008 | 0.352 | <0.001 |
| $NO_x$ | 0.132 | 0.102 | 0.286 | <0.001 |
| $NO_2/NO$ | 0.001 | 0.986 | 0.162 | 0.048 |


Table 2 Correlations of BSOA with sulfate and $O_x$

| | Sulfate | | | $O_x$ | | |
|---|---|---|---|---|---|---|
| | Slope | $p$-value | % [a] | Slope | $p$-value | % [a] |
| $SOA_M$ | 0.112 | <0.001 | 45 | 0.013 | <0.001 | 57 |
| $SOA_I$ | 0.020 | <0.001 | 34 | 0.003 | <0.001 | 50 |
| $SOA_C$ | 0.041 | <0.001 | 46 | 0.004 | <0.001 | 55 |
| BSOA | 0.172 | <0.001 | 44 | 0.019 | <0.001 | 55 |

[a] Percentages of SOA reduction at 50% decline of sulfate or $O_x$