# Peer review of "Impact of anthropogenic emissions on biogenic secondary"

_Atmospheric Chemistry and Physics, 2019_

## Referee Comment (RC1) · Anonymous Referee #1 · 13 Jul 2019

Review of "Impact of anthropogenic emissions on biogenic secondary organic aerosol: Observation in the Pearl River Delta, South China" by Zhang et al.

This manuscript presents the annual variations of SOA tracers from biogenic VOCs at nine sites in PRD region. The measured biogenic SOA tracers are found to be correlated with Ox and anthropogenic sulfate, indicating the impacts of anthropogenic emissions on biogenic SOA formation. This is an extensive study by analyzing 170 filters. Overall, the data analysis is solid and the manuscript is well-written. I recommend publication after major revisions.

Major Comments

1.      Recent studies [1-2] demonstrated that C5-alkene triols and 3-methyltetrahydrofuran-3,4-diols are largely GC/EI-MS artifacts from the degradation of methyltetrol sulfates and dimers. The authors used figure 7 (ternary plots) to argue that these tracers are indeed formed from different pathways rather than thermal decomposition. However, I beg to differ. The lack of correlation between IEPOX-derived SOA tracers and be explained by that the three tracers in figure 7 arise from the thermal decomposition of different dimers/OS and the parent dimers/OS concentration varies with sites and season. To fully prove that the three tracers are not decomposition products, the authors need to sample authentic methyltetrol sulfate standard with GC-MS.

2.      The authors use the BSOA vs sulfate slope to infer the magnitude of sulfate control on BSOA. As the authors have performed the same measurements in the same region for a long time, I encourage the authors to look at their historic measurements, based on which to estimate the sulfate-control magnitude. As shown in figure 1, the O3 concentration has been relatively flat in the past 13 years, but SO2 concentration has largely declined. This provides a nice opportunity to deconvolve the effect of sulfate vs O3 on BSOA formation.

Minor Comments

1.      Line 34. Replace "high-generation" with "later-generation" throughout the manuscript including acronyms.

2.      Line 231-232. The correlation between $SOA_{M\_H}$ and sulfate is intriguing. The formation of $SOA_{M\_H}$ tracers, like MBTCA, does not involve sulfate. Thus, how to explain the correlation?

3.      Please show the correlation of isoprene SOA tracers with sulfate and Ox, like figure 5.

4.      Figure 5. With so many data on the plots, it is difficult to examine the correlation at each site. I suggest to make a scatter plot for each site and then synthesize a figure like figure 4.

5.      The correlation between b-caryophyllenic acid (CA) and levoglucosan is interesting. Does biomass burning emit CA?

6.      Line 331. The authors need to be careful about the salting-in effect, because it is highly compound-specific. Xu et al. 2015 proposed that sulfate introduces salting-in effect on IEPOX, but this is just a hypothesis. It would be overreaching to argue that sulfate has salting-in effect on beta-caryophyllene SOA.

Reference

1.      Cui, T.; Zeng, Z.; dos Santos, E. O.; Zhang, Z.; Chen, Y.; Zhang, Y.; Rose, C. A.; Budisulistiorini, S. H.; Collins, L. B.; Bodnar, W. M., et al. Development of a Hydrophilic Interaction Liquid Chromatography (Hilic) Method for the Chemical Characterization of Water-Soluble Isoprene Epoxydiol (Iepox)-Derived Secondary Organic Aerosol. *Environmental Science: Processes & Impacts* **2018**.
2.      Watanabe, A. C.; Stropoli, S. J.; Elrod, M. J. Assessing the Potential Mechanisms of Isomerization Reactions of Isoprene Epoxydiols on Secondary Organic Aerosol. *Environ Sci Technol* **2018,** *52*, 8346-8354.

---

## Referee Comment (RC2) · Anonymous Referee #2 · 21 Jul 2019

This manuscript by Zhang et al. represent a detailed analysis on tracer organic compounds quantified in PM2.5 samples collected at 23sites in the Pearl River Delta (PRD) region. Based on the tracer concentrations, the authors performed correlation analyses and source apportionment to understand the source of secondary organic aerosol (SOA), as well as the impact of anthropogenic emissions to biogenic SOA (BSOA). The topic of the study is timely and is within the scope of ACP. Especially, the interaction of anthropogenic and biogenic emissions in relatively polluted regions, such as the PRD, is not well understood, and the results from this study is highly valuable. The manuscript is of high quality in terms of chemical analysis, discussion, implication, and literary presentation. I recommend publication of this work in ACP and I have a number of minor and technical suggestions:

- Section 2.2: The authors quantified quite a number of organic tracers. The authors should justify how representative are these tracers for $SOA_I$, $SOA_M$, and $SOA_C$. In particular, I am not familiar with HDMGA and HGA as tracers for monoterpenes. Citation is needed to justify the specificity and selectivity of these tracers.
- Section 2.3: It seems that the recovery for erythritol is low. Why it is low and how is the result of the recovery test reflected in the quantification of related tracers?
- The authors use $Ox$ as an indicator of the atmospheric oxidative capacity. However, caution is needed, as $Ox = (O_3 + NO_2)$ represents the total $O_3$. While $O_3$ is certainly an important oxidant, the contribution of OH radical (which is perhaps more important) is not considered in $Ox$. Is there any evidence showing that OH concentration is also high when the $Ox$ concentration is high?
- Section 3.2.3: It is interesting that β-caryophyllenic acid (CA) was observed to be high in the winter and correlate with BB tracers. Can the authors comment on whether this observation place question on the selectivity and specificity of CA as a tracer for SOAc?
- Related to CA, I have come across compounds that have very similar names to β-caryophillenic acid, namely, β-caryophyllinic acid and β-caryophyllonic acid. (Jaoui et al. (2007) Geophys. Res. Lett.; Bé et al. (2019) ACS Earth Space Chem.). Is β-caryophyll**e**nic acid measured in this study a different compound or this is simply a typo?
- Section 3.3: The impact of anthropogenic emissions to BSOA is perhaps one of the most important implications in this manuscript, but the current discussion appears weak. Based on the slopes obtained from the correlation studies, the authors implies that reducing $SO_4$ and $Ox$ in the atmosphere will lead to reduction of BSOA. However, this type of correlation analysis exhibits only correlation but not causation. The authors should justify why reducing anthropogenic emissions can likely reduce BSOA. One way to do this, I think, is to add more discussion on the mechanisms behind the influence of anthropogenic emissions to BSOA.

---

## Author Comment (AC1) · 8 Oct 2019

**Response to the reviewer's comments**

**"Impact of anthropogenic emissions on biogenic secondary organic aerosol: Observation in the Pearl River Delta, South China"** *by* **Yu-Qing Zhang et al.**

In the following, the comments made by the referees appear in black, while our replies are in blue, and the revised texts in the manuscript are in red.

**Reviewer #1**

This manuscript presents the annual variations of SOA tracers from biogenic VOCs at nine sites in PRD region. The measured biogenic SOA tracers are found to be correlated with $O_x$ and anthropogenic sulfate, indicating the impacts of anthropogenic emissions on biogenic SOA formation. This is an extensive study by analyzing 170 filters. Overall, the data analysis is solid and the manuscript is well-written. I recommend publication after major revisions.

We thank the reviewer for the helpful comments and suggestions for this manuscript. We respond to the reviewer point by point below and modified the manuscript according to the comments.

**Major Comments**

1. Recent studies [1-2] demonstrated that $C_5$-alkene triols and 3-methyltetrahydrofuran-3,4-diols are largely GC/EI-MS artifacts from the degradation of methyltetrol sulfates and dimers. The authors used figure 7 (ternary plots) to argue that these tracers are indeed formed from different pathways rather than thermal decomposition. However, I beg to differ. The lack of correlation between IEPOX-derived SOA tracers and be explained by that the three tracers in figure 7 arise from the thermal decomposition of different dimers/OS and the parent dimers/OS concentration varies with sites and season. To fully prove that the three tracers are not decomposition products, the authors need to sample authentic methyltetrol sulfate standard with GC-MS.

Reply: We agree that thermal degradation of methyltetrol sulfates and dimers could produce $C_5$-alkene triols, 3-methyltetrahydrofuran-3,4-diols and 2-methyltetrols during the GC/MS analysis (Watanabe et al., 2018). Using a hydrophilic interaction liquid chromatography (HILIC) method developed by Jason

Surratt group, Cui et al (2018) estimated that 30.0%, 42.8%, and 14.7% of $C_5$-alkene triols measured by

GC/MS were attributed to the potential thermal degradation of the 2-methyltetrol sulfates in the SOA

from β-IEPOX, and the $PM_{2.5}$ samples from Look Rock and Manaus sites, respectively. The fractions of

2-methyltetrols attributed to thermal degradation were 11.1%. And approximately all

3-MeTHF-3,4-diols were produced from thermal degradation. Recently, we also measured

2-methyltetrol sulfates in two samples at HS and TS sites, respectively (see Table R1-1 below).

Assuming that all the 2-methyltetrol sulfates decomposed to these tracers, the thermal decomposition of

2-methyltetrol sulfates would account for 15.1-31.6% of $C_5$-alkene triols, 6.0-10.0% of 2-methyltetrols and all 3-methyltetrahydrofuran-3,4-diols measured by GC/ MS. Thus, $C_5$-alkene triols and

2-methyltetrols are major from isoprene oxidation rather than thermal decomposition of 2-methyltetrol sulfates, while 3-methyltetrahydrofuran-3,4-diols are only in trace amount in the air and might be produced largely from thermal degradation. Coupled with significant variations in tracer compositions observed in the PRD, we believe that these $SOA_I$ tracers are mainly formed through different pathways in the ambient atmosphere, although part of them might arise from the thermal decomposition of different dimers/OSs and the parent dimers/OSs varies with sites and seasons.

All these discussion (see below) are added in the revised manuscript in line 250-271.

"Previous studies found that thermal decomposition of low volatility organics in IEPOX-derived

SOA could produce $SOA_I$ tracers, e.g. 2-MTLs, $C_5$-alkene triols and 3-MeTHF-3,4-diols (Lopez-Hilfiker et al., 2016, Watanabe et al., 2018). This means that these tracers detected by GC-MSD

might be generated from thermal decomposition of IEPOX-derived SOA. As estimated by Cui et al.

(2018), 14.7-42.8% of $C_5$-alkene triols, 11.1% of 2-MTLs and approximately all

3-MeTHF-3,4-diols.measured by GC/ MSD could be attributed to the thermal degradation of

2-MTLs-derived organosulfates (MTL-OSs). We also measured MTL-OSs in two samples at HS and

TS sites, respectively (Table S6) using the widely used LC-MS approach (He et al., 2014, 2018).

Assuming that all MTL-OSs decomposed to these tracers, the thermal decomposition of MTL-OSs would account for 15.1-31.6% of $C_5$-alkene triols, 6.0-10.0% of 2-MTLs and all 3-MeTHF-3,4-diols measured by GC/ MSD. Thus, $C_5$-alkene triols and 2-MTLs are major from isoprene oxidation rather than thermal decomposition of MTL-OSs, while 3-MeTHF-3,4-diols are only in trace amount in the air and might be produced largely from thermal degradation.

Moreover, we see significant variations in $SOA_I$ tracer compositions in the PRD. For instant,

C$_5$-alkene triols have three isomers. If these tracers were mainly generated from a thermal process, their compositions should be similar in different samples. In fact, the relative abundances of three C$_5$-alkene triol isomers significantly changed from site to site (Figure 7) and season to season (Figure S8), and their compositions in the PRD were different from those measured in the chamber samples (Lin et al., 2012). In addition, the slopes of linear correlations among these IEPOX-derived SOA tracers also varied from site to site (Figure S9). Coupled with the seasonal trend of 2-MGA/2-MTLs ratios, the apparent variations in SOA$_I$ tracer compositions demonstrate that these SOA$_I$ tracers are mainly formed through different pathways in the ambient atmosphere, although part of them might arise from the thermal decomposition of different dimers/OSs and the parent dimers/OSs varies with sites and seasons."

Table R1-1 Concentrations of isoprene SOA products at HS and TS sites

|  | HS 20150701 | TS 20150701 |
|---|---|---|
| 2-Methyltetrol sulfates (ng m$^{-3}$) | 6.65 | 2.99 |
| C$_5$-alkene triols (ng m$^{-3}$) | 11.5 | 10.8 |
| 2-Methyltetrols (ng m$^{-3}$) | 41.8 | 31.2 |
| 3-MeTHF-3,4-diols (ng m$^{-3}$) | 0.482 | 0.227 |

Reference
1. Cui, T., Zeng, Z., dos Santos, E. O., Zhang, Z., Chen, Y., Zhang, Y., Rose, C. A., Budisulistiorini, S. H., Collins, L. B., Bodnar, W. M., de Souza, R. A. F., Martin, S. T., Machado, C. M. D., Turpin, B. J., Gold, A., Ault, A. P., and Surratt, J. D.: Development of a hydrophilic interaction liquid chromatography (HILIC) method for the chemical characterization of water-soluble isoprene epoxydiol (IEPOX)-derived secondary organic aerosol, Environ. Sci.: Processes Impacts, 20, 1524-1536, 10.1039/C8EM00308D, 2018.
2. Watanabe, A. C., Stropoli, S. J., and Elrod, M. J.: Assessing the potential mechanisms of isomerization reactions of isoprene epoxydiols on secondary organic aerosol, Environ. Sci. Technol., 52, 8346-8354, 10.1021/acs.est.8b01780, 2018.

2. The authors use the BSOA vs sulfate slope to infer the magnitude of sulfate control on BSOA. As the authors have performed the same measurements in the same region for a long time, I encourage the authors to look at their historic measurements, based on which to estimate the sulfate-control magnitude. As shown in figure 1, the O$_3$ concentration has been relatively flat in the past 13 years, but SO$_2$ concentration has largely declined. This provides a nice opportunity to deconvolve the effect of sulfate vs O$_3$ on BSOA formation.

Reply: Thanks for the suggestion. Since 2007, we have carried out one-month campaign during fall-winter season every year at a regional site, Wanqingsha (WQS) in the PRD. At present, we have long-term data of sulfate from 2007 to 2016. Similar to $SO_2$, sulfate keeps decreasing in the past decade (see Figure R1-1 below). Unfortunately, we have not completed the time-consuming measurements of BSOA tracers yet. Currently, we cannot deconvolve the effect of sulfate vs $O_3$ on BSOA formation based on long-term data. We will write another manuscript focusing on these long-term measurements and discussing the changing effects of sulfate and $O_3$ on BSOA formation.

Instead, we compared the data in 2015 with those during fall-winter season in 2008 at WQS (Ding et al., 2012), since BSOA tracers, sulfate and $O_x$ were all measured in both studies. As the below Table R1-2 showed, all BSOA species positively correlated with sulfate but exhibited no $O_x$ dependence. Thus, BSOA formation in 2008 was largely influenced by sulfate, probably due to high sulfate levels then (as high as 46.8 μg m$^{-3}$). Coupled with the decrease trend of sulfate and the relatively flat trend of $O_x$, such a difference in sulfate and $O_x$ dependence between 2015 and 2008 highlights the critical role of $O_x$ in BSOA formation currently in the PRD.

In the revised manuscript, we add these discussion in line 324-332 "We further compared the results in 2015 with those during fall-winter season in 2008 at WQS (Ding et al., 2012). We found that all BSOA species positively correlated with sulfate but exhibited no $O_x$ dependence (Table S7). Thus, in 2008 BSOA formation was largely influenced by sulfate, probably due to high sulfate levels then (as high as 46.8 μg m$^{-3}$). Owing to strict control of $SO_2$ emissions (Wang et al., 2013), ambient $SO_2$ significantly shrank over the PRD (Figure 1b). Our long-term observation during fall-winter season at WQS also witnessed a decreasing trend of sulfate from 2007 to 2016 (Figure S10). However, $O_x$ levels did not decrease during the past decade (Figure 1b) and $O_x$ concentrations were much higher than sulfate in 2015 in the PRD (96.1 ± 14.9 μg m$^{-3}$ vs. 8.44 ± 1.09 μg m$^{-3}$ on average). All these underline the importance of $O_x$ in BSOA formation currently in the PRD."

[Figure]

Figure R1-1 Long-term variation of sulfate in fall at WQS from 2007-2016.

Table R1-2 Correlations of BSOA with sulfate and $O_x$ during fall-winter in 2008 at WQS

| | Sulfate (2008-PRD) | | | $O_x$ (2008-PRD) | |
| --- | --- | --- | --- | --- | --- |
| | Slope | *p*-value | % [a] | Slope | *p*-value |
| $SOA_M$ | 0.023 | 0.005 | 50 | - | 0.551 |
| $SOA_I$ | 0.032 | <0.001 | 76 | - | 0.509 |
| $SOA_C$ | 0.032 | <0.001 | 87 | - | 0.139 |
| BSOA | 0.087 | <0.001 | 69 | - | 0.563 |

[a] Percentages of SOA reduction at 50% decline of sulfate or $O_x$

Reference

Ding, X., Wang, X., Gao, B., Fu, X., He, Q., Zhao, X., Yu, J., and Zheng, M.: Tracer based estimation of secondary organic carbon in the Pearl River Delta, South China, J. Geophys. Res-Atmos., 117, D05313, 10.1029/2011JD016596, 2012.

**Minor Comments**

1. Line 34. Replace "high-generation" with "later-generation" throughout the manuscript including acronyms.

Reply: We have replaced it as suggested. We also replace "$SOA_{M\_H}$" with "$SOA_{M\_L}$" throughout the manuscript.

2. Line 231-232. The correlation between $SOA_{M\_H}$ and sulfate is intriguing. The formation of $SOA_{M\_H}$

tracers, like MBTCA, does not involve sulfate. Thus, how to explain the correlation?

Reply: Müller et al. (2012) reported that MBTCA could be formed through the gas-phase OH oxidation of pinonic acid. The triacid nature of MBTCA makes it highly water-soluble and able to partition into cloud water and aerosol liquid water (Aljawhary et al 2016). Besides the gas-phase OH oxidation, the heterogeneous OH oxidation of pinonic acid could also produce MBTCA (Lai et al. 2015). Aljawhary et al. (2016) reported the kinetics and mechanism of pinonic acid oxidation in acidic solutions and found that the molar yields of MBTCA through the aqueous-phase reactions were similar to those in the gas-phase oxidation. Sulfate is a key species in particles that determines aerosol liquid water amount, aerosol acidity, and particle surface area (Xu et al., 2015, 2016). Thus, the increase of sulfate could promote aqueous and heterogeneous reactions and produce substantial MBTCA in particles.

In the revised manuscript, we add these discussion in line 187-197 "On the other hand, sulfate is a key species in particles that determines aerosol liquid water amount, aerosol acidity, and particle surface area (Xu et al., 2015, 2016). Thus, the increase of sulfate could promote aqueous and heterogeneous reactions. In this study, the $SOA_{M\_F}$ tracers poorly correlated with sulfate (Figure 5c), while the $SOA_{M\_L}$ tracers positively correlated with sulfate at all the 9 sites (Figure 5d). At each site the

$SOA_{M\_L}$ tracers exhibited more sulfate dependence than $SOA_{M\_F}$ tracers (Figure S5). This suggested that sulfate also played a critical role in forming $SOA_{M\_L}$ tracers through the particle-phase reactions.

Besides the gas-phase OH oxidation (Müller et al., 2012), the heterogeneous OH oxidation of pinonic acid could also produce $SOA_{M\_L}$ tracers (Lai et al. 2015). Aljawhary et al., (2016) reported the kinetics and mechanism of pinonic acid oxidation in acidic solutions and found that the molar yields of

MBTCA through the aqueous-phase reactions were similar to those in the gas-phase oxidation."

Reference
1. Aljawhary, D., Zhao, R., Lee, A. K. Y., Wang, C., and Abbatt, J. P. D.: Kinetics, mechanism, and
secondary organic aerosol yield of aqueous phase photo-oxidation of α-pinene oxidation products, J.
Phys. Chem. A., 120, 1395-1407, 10.1021/acs.jpca.5b06237, 2016.
2. Lai, C., Liu, Y., Ma, J., Ma, Q., Chu, B., and He, H.: Heterogeneous kinetics of cis-pinonic acid with
hydroxyl radical under different environmental conditions, J. Phys. Chem. A., 119, 6583-6593,
10.1021/acs.jpca.5b01321, 2015.
3. Müller, L., Reinnig, M. C., Naumann, K. H., Saathoff, H., Mentel, T. F., Donahue, N. M., and

Hoffmann, T.: Formation of 3-methyl-1,2,3-butanetricarboxylic acid via gas phase oxidation of pinonic
acid – a mass spectrometric study of SOA aging, Atmos. Chem. Phys., 12, 1483-1496,
10.5194/acp-12-1483-2012, 2012.
4. Xu, L., Guo, H., Boyd, C. M., Klein, M., Bougiatioti, A., Cerully, K. M., Hite, J. R.,
Isaacman-VanWertz, G., Kreisberg, N. M., Knote, C., Olson, K., Koss, A., Goldstein, A. H., Hering, S.
V., de Gouw, J., Baumann, K., Lee, S.-H., Nenes, A., Weber, R. J., and Ng, N. L.: Effects of
anthropogenic emissions on aerosol formation from isoprene and monoterpenes in the southeastern
United States, P. Natl. Acad. Sci. USA., 112, 37-42, 10.1073/ P.Natl.Acad.Sci.USA.1417609112, 2015.
5. Xu, L., Middlebrook, A. M., Liao, J., de Gouw, J. A., Guo, H., Weber, R. J., Nenes, A.,
Lopez-Hilfiker, F. D., Lee, B. H., Thornton, J. A., Brock, C. A., Neuman, J. A., Nowak, J. B., Pollack, I.
B., Welti, A., Graus, M., Warneke, C., and Ng, N. L.: Enhanced formation of isoprene-derived organic
aerosol in sulfur-rich power plant plumes during Southeast Nexus, J. Geophys. Res-Atmos., 121,
11137-11153, 10.1002/2016JD025156, 2016.

3. Please show the correlation of isoprene SOA tracers with sulfate and $O_x$, like figure 5.

Reply: As suggested, we checked the correlations of isoprene SOA tracers with sulfate and $O_x$ (see below Figure R1-2) and added the figure into the Supporting Information as Figure S6. The

$NO/NO_2$-channel product exhibited more $O_x$ and sulfate dependence than $HO_2$-channel products. In the revised manuscript, we add these discussion in line 234-236 "We also checked the correlations of $SOA_I$

tracers with $O_x$ and sulfate (Figure S6). The $NO/NO_2$-channel product exhibited more $O_x$ and sulfate dependance than $HO_2$-channel products."

[Figure]

Figure R1-2 Correlations of SOA_{HO2-channel} tracers and SOA_{NO/NO2-channel} tracers with $O_x$ (a, b) and sulfate (c, d)

4. Figure 5. With so many data on the plots, it is difficult to examine the correlation at each site. I

suggest to make a scatter plot for each site and then synthesize a figure like figure 4.

Reply: As suggested, we add the scatter plots of SOA_{M_F} tracers and SOA_{M_L} tracers with $O_x$ and sulfate at each site in Supporting Information of the revised manuscript (Figure S4 and S5, see the

Figure R1-3, R1-4 below).

[Figure]

179 Figure R1-3 Correlations of SOA$_{M\_F}$ tracers and SOA$_{M\_L}$ tracers with O$_x$

[Figure]

Figure R1-4 Correlations of SOA$_{M\_F}$ tracers and SOA$_{M\_L}$ tracers with sulfate

5. The correlation between b-caryophyllenic acid (CA) and levoglucosan is interesting. Does biomass burning emit CA?

Reply: To the best of our knowledge, biomass burning does not emit CA but its precursor. Sesquiterpenes, including β-caryophyllene are synthesized and stored in plant tissues partly to protect plants from insects and pathogens (Keeling and Bohlmann, 2006). When biomass burning evens happen, high temperature could release substantial sesquiterpenes into the air (Ciccioli et al. 2014). One the other hand, the oxidation of β-caryophyllene by the OH radical and $O_3$ is very rapid. Under typical oxidation conditions in the air of PRD, the lifetimes of β-caryophyllene are only several minutes. This means that once emitted from biomass burning, β-caryophyllene could react rapidly and form CA immediately. Thus, it is expected to see a positive correlation between CA and levoglucosan.

In the revised manuscript, we add this discussion in Line 285-288 "The oxidation of β-caryophyllene by the OH radical and $O_3$ is very rapid. Under typical oxidation conditions in the air of PRD, the lifetimes of β-caryophyllene are only several minutes (Table S5). Once emitted from vegetation or biomass burning, β-caryophyllene will react rapidly and form CA immediately. This partly explains the positive correlations between CA and levoglucosan in the PRD."

Reference

1. Keeling, C. I., and Bohlmann, J.: Genes, enzymes and chemicals of terpenoid diversity in the constitutive and induced defence of conifers against insects and pathogens, New Phytol., 170, 657-675, 10.1111/j.1469-8137.2006.01716.x, 2006.
2. Ciccioli, P., Centritto, M., and Loreto, F.: Biogenic volatile organic compound emissions from vegetation fires, Plant Cell Environ., 37, 1810-1825, 10.1111/pce.12336, 2014.

6. Line 331. The authors need to be careful about the salting-in effect, because it is highly compound-specific. Xu et al. 2015 proposed that sulfate introduces salting-in effect on IEPOX, but this is just a hypothesis. It would be overreaching to argue that sulfate has salting-in effect on beta-caryophyllene SOA.

Reply: We agree. In the revised manuscript, we change the statement as "In addition, the increase of sulfate could raise aerosol acidity and thereby promote aqueous and heterogeneous reactions to form $SOA_C$." in line 290-291.

---

## Author Comment (AC2) · 8 Oct 2019

**Response to the reviewer's comments**

**"Impact of anthropogenic emissions on biogenic secondary organic aerosol: Observation in the Pearl River Delta, South China"** *by* **Yu-Qing Zhang et al.**

In the following, the comments made by the referees appear in black, while our replies are in blue, and the revised texts in the manuscript are in red.

**Reviewer #2**

This manuscript by Zhang et al. represent a detailed analysis on tracer organic compounds quantified in $PM_{2.5}$ samples collected at 23 sites in the Pearl River Delta (PRD) region. Based on the tracer concentrations, the authors performed correlation analyses and source apportionment to understand the source of secondary organic aerosol (SOA), as well as the impact of anthropogenic emissions to biogenic SOA (BSOA). The topic of the study is timely and is within the scope of ACP. Especially, the interaction of anthropogenic and biogenic emissions in relatively polluted regions, such as the PRD, is not well understood, and the results from this study is highly valuable. The manuscript is of high quality in terms of chemical analysis, discussion, implication, and literary presentation. I recommend publication of this work in ACP and I have a number of minor and technical suggestions.

We thank the referee for the positive evaluation of our manuscript and the useful comments and suggestions. A point-by-point response is included below and we have revised the manuscript according to the comments.

1. - Section 2.2: The authors quantified quite a number of organic tracers. The authors should justify how representative are these tracers for $SOA_I$, $SOA_M$, and $SOA_C$. In particular, I am not familiar with HDMGA and HGA as tracers for monoterpenes. Citation is needed to justify the specificity and selectivity of these tracers.

Reply: For $SOA_I$ tracers, Claeys et al (2004a) first identified 2-methyltetrols in Amazon aerosols and disclosed the importance of $SOA_I$. They further identified 2-methylglyceric acid (Claeys et al., 2004b) and $C_5$-alkene triols (Wang et al., 2005) as specific markers for acid-catalyzed ring opening of the isoprene-derived epoxides (e.g. MAE/HMML and IEPOXs). Lin et al., (2012) identified

3-MeTHF-3,4-diols as the products of acid-catalyzed intermolecular rearrangement of IEPOX on acidic particles.

For SOA$_M$ tracers, pinic acid and pinonic acid were firstly identified in the chamber-generated

SOA from the ozonolysis and OH oxidation of pinenes (Christoffersen et al 1998). Further oxidation of pinic acid and pinonic acid can form highly oxidized products, e.g. MBTCA whose chemical structure was finally identified by Szmigielski et al. (2007). 3-Hydroxyglutaric acid (HGA) and

3-hydroxy-4,4-dimethylglutaric acid (HDMGA) were first reported as SOA$_M$ tracers (U1 and U2

compounds in the Figure R2-1 below) by Claeys et al (2007). As a tracer for SOA$_C$, β-Caryophyllenic acid was first identified by Jaoui et al. (2007).

In the revised manuscript, we add all these references to justify the specificity and selectivity of these tracers in Line 107-116.

[Figure]

Figure R2-1 GC/MS TICs obtained for the trimethylsilylated extract of PM$_{2.5}$ aerosols and the chemical
structures of HGA (U1) and HDMGA (U2) from Claeys et al (2007).

Reference

1. Christoffersen, T. S., Hjorth, J., Horie, O., Jensen, N. R., Kotzias, D., Molander, L. L., Neeb, P.,

Ruppert, L., Winterhalter, R., Virkkula, A., Wirtz, K., and Larsen, B. R.: Cis-pinic acid, a possible precursor for organic aerosol formation from ozonolysis of α-pinene, Atmos. Environ., 32, 1657-1661, https://doi.org/10.1016/S1352-2310(97)00448-2, 1998.

2. Claeys, M., Graham, B., Vas, G., Wang, W., Vermeylen, R., Pashynska, V., Cafmeyer, J., Guyon, P., Andreae, M. O., Artaxo, P., and Maenhaut, W.: Formation of secondary organic aerosols through photooxidation of isoprene, Science, 303, 1173-1176, 10.1126/science.1092805, 2004a.

3. Claeys, M., Wang, W., Ion, A. C., Kourtchev, I., Gelencsér, A., and Maenhaut, W.: Formation of secondary organic aerosols from isoprene and its gas-phase oxidation products through reaction with hydrogen peroxide, Atmos. Environ., 38, 4093-4098, https://doi.org/10.1016/j.atmosenv.2004.06.001, 2004b.

4. Claeys, M., Szmigielski, R., Kourtchev, I., Van der Veken, P., Vermeylen, R., Maenhaut, W., Jaoui, M., Kleindienst, T. E., Lewandowski, M., Offenberg, J. H., and Edney, E. O.: Hydroxydicarboxylic acids: Markers for secondary organic aerosol from the photooxidation of α-pinene, Environ. Sci. Technol., 41, 1628-1634, 10.1021/es0620181, 2007.

5. Jaoui, M., Lewandowski, M., Kleindienst, T. E., Offenberg, J. H., and Edney, E. O.: β-caryophyllinic acid: An atmospheric tracer for β-caryophyllene secondary organic aerosol, Geophys. Res. Let., 34, 10.1029/2006gl028827, 2007.

6. Lin, Y.-H., Zhang, Z., Docherty, K. S., Zhang, H., Budisulistiorini, S. H., Rubitschun, C. L., Shaw, S., Knipping, E., Edgerton, E. S., Kleindienst, T. E., Gold, A., and Surratt, J. D.: Isoprene epoxydiols as precursors to secondary organic aerosol formation: Acid-catalyzed reactive uptake studies with authentic standards, Environ. Sci. Technol., 46, 189-195, 10.1021/es202554c, 2012.

7. Szmigielski, R., Surratt, J. D., Gómez-González, Y., Van der Veken, P., Kourtchev, I., Vermeylen, R., Blockhuys, F., Jaoui, M., Kleindienst, T. E., Lewandowski, M., Offenberg, J. H., Edney, E. O., Seinfeld, J. H., Maenhaut, W., and Claeys, M.: 3-methyl-1,2,3-butanetricarboxylic acid: An atmospheric tracer for terpene secondary organic aerosol, Geophys. Res. Let., 34, 10.1029/2007gl031338, 2007.

8. Wang, W., Kourtchev, I., Graham, B., Cafmeyer, J., Maenhaut, i., and Claeys, M.: Characterization of oxygenated derivatives of isoprene related to 2-methyltetrols in Amazonian aerosols using trimethylsilylation and gas chromatography/ion trap mass spectrometry, Rapid Commun. Mass Spectrom., 19, 1343-1351, 10.1002/rcm.1940, 2005.

2. - Section 2.3: It seems that the recovery for erythritol is low. Why it is low and how is the result of the recovery test reflected in the quantification of related tracers?

Reply: In our study, apart from using internal standards, we also tested "absolute" recoveries for the standard compounds by analyzing spiked samples. The recovery results in QAQC showed the absolute recovery of each standard compound. Compared with PNA and octadecanoic acid, erythritol has the smallest carbon number and the lightest molecular weight. The loss of erythritol during chemical analysis should be highest among the three target compounds. Thus, the absolute recovery of erythritol is low.

For tracer quantification, we added internal standards to each sample before extraction and quantified SOA tracers using the internal standards approach. The internal calibration procedure uses the peak area ratio of target compound to internal standard to do the quantification. Since internal standards have similar chemical structure and/or retention time to the target compounds, their loss should be comparable to those of target compounds during sample analysis. The internal standard calibration based on peak area ratios is in fact already recovery corrected with the assumption that internal standards and target compounds have identical recoveries. Thus, the low absolute recovery of erythritol does not affect the quantification of related tracers.

3. - The authors use $O_x$ as an indicator of the atmospheric oxidative capacity. However, caution is needed, as $O_x = (O_3 + NO_2)$ represents the total $O_3$. While $O_3$ is certainly an important oxidant, the contribution of OH radical (which is perhaps more important) is not considered in $O_x$. Is there any evidence showing that OH concentration is also high when the $O_x$ concentration is high?

Reply: We admit that high $O_x$ itself cannot indicate high oxidative capacity. In fact, Hofzumahaus et al., (2009) observed extremely high OH concentrations ($15 \times 10^6$ cm$^{-3}$ around noon) in the PRD and proposed a recycling mechanism which increases the stability of OH in the air of polluted regions. Since $O_3$, $NO_x$ and OH are intimately linked in atmospheric chemistry, we think that the atmospheric oxidative capacity keeps high in the PRD. Because we did not have OH measurements in the current study, in the revised manuscript we remove the statement "$O_x$ as an indicator of the atmospheric oxidative capacity" and change the related discussion as:

"The SOA$_C$ tracer elevated in winter at all sites and positively correlated with levoglucosan, $O_x$, and sulfate. Thus, the unexpected increase of SOA$_C$ in wintertime might be highly associated with the enhancement of biomass burning, $O_3$ chemistry and sulfate component in the PRD." (Line 31-34)

"However, $O_3$ and oxidant ($O_x$, $O_x = O_3 + NO_2$) are still in high levels and do not decrease apparently (Figure 1b). Hofzumahaus et al., (2009) observed extremely high OH concentrations in the PRD and proposed a recycling mechanism which increases the stability of OH in the air of polluted regions. All these indicate high atmospheric oxidative capacity in the PRD, since $O_3$, $NO_x$ and OH are intimately linked in atmospheric chemistry." (Line 64-68)

"Due to the compromise of opposite seasonal trends of $O_3$ and $NO_2$, $O_x$ showed less seasonal variation (Figure 3f) compared with other gaseous pollutants. And annual-mean $O_x$ reached $96.1 \pm 14.9$ μg m$^{-3}$. These indicated significant $O_3$ pollution all the year in the PRD." (Line 147-150)

Reference
Hofzumahaus, A., Rohrer, F., Lu, K., Bohn, B., Brauers, T., Chang, C.-C., Fuchs, H., Holland, F., Kita,
K., Kondo, Y., Li, X., Lou, S., Shao, M., Zeng, L., Wahner, A., and Zhang, Y.: Amplified trace gas
removal in the troposphere, Science, 324, 1702-1704, 10.1126/science.1164566, 2009.

4. - Section 3.2.3: It is interesting that β-caryophyllenic acid (CA) was observed to be high in the winter and correlate with BB tracers. Can the authors comment on whether this observation place question on the selectivity and specificity of CA as a tracer for SOAc ?

Reply: β-Caryophyllenic acid (CA) was identified in SOA produced through the ozonolysis and photooxidation of β-caryophyllene (Jaoui et al. 2007). Previous studies have demonstrate that biomass burning does emit substantial sesquiterpenes (Ciccioli et al. 2014; Mentel et al., 2013) which are synthesized and stored in plant tissues to protect plants from insects and pathogens (Keeling and

Bohlmann, 2006). One the other hand, the oxidation of β-caryophyllene in the air is very rapid. Under typical OH and $O_3$ levels in the air of PRD, the lifetimes of β-caryophyllene are only several minutes.

Once emitted from biomass burning, β-caryophyllene could react rapidly and form CA immediately.

Thus, it is expected to see a positive correlation between CA and levoglucosan. The unexpected high levels of CA in the winter indicated that biomass burning could be an important source of $SOA_C$ in the

PRD, especially in wintertime.

In the revised manuscript, we add this discussion in Line 285-290 "The oxidation of

β-caryophyllene by the OH radical and $O_3$ is very rapid. Under typical oxidation conditions in the air of

PRD, the lifetimes of β-caryophyllene are only several minutes (Table S5). Once emitted from vegetation or biomass burning, β-caryophyllene will react rapidly and form CA immediately. This partly explains the positive correlations between CA and levoglucosan in the PRD. The unexpected high levels of CA in the winter indicated that biomass burning could be an important source of $SOA_C$

in the PRD, especially in wintertime."

Reference
1. Ciccioli, P., Centritto, M., and Loreto, F.: Biogenic volatile organic compound emissions from
vegetation fires, Plant Cell Environ., 37, 1810-1825, 10.1111/pce.12336, 2014.
2. Jaoui, M., Lewandowski, M., Kleindienst, T. E., Offenberg, J. H., and Edney, E. O.: β-caryophyllinic
acid: An atmospheric tracer for β-caryophyllene secondary organic aerosol, Geophys. Res. Let., 34,
10.1029/2006gl028827, 2007..
3. Keeling, C. I., and J. Bohlmann: Genes, enzymes and chemicals of terpenoid diversity in the
constitutive and induced defence of conifers against insects and pathogens, New Phytol., 170(4), 657–

675, 2006.

4. Mentel, T. F., Kleist, E., Andres, S., Maso, M. D., Hohaus, T., Kiendler-Scharr, A., Rudich, Y.,

Springer, M., Tillmann, R., Uerlings, R., Wahner, A., and Wildt, J.: Secondary aerosol formation from stress-induced biogenic emissions and possible climate feedbacks, Atmos. Chem. Phys., 13, 8755-8770,

10.5194/acp-13-8755-2013, 2013.

5. - Related to CA, I have come across compounds that have very similar names to β-caryophillenic acid, namely, β-caryophyllinic acid and β-caryophyllonic acid. (Jaoui et al. (2007) Geophys. Res. Lett.;

Bé et al. (2019) ACS Earth Space Chem.). Is β-caryophyllenic acid measured in this study a different compound or this is simply a typo?

Reply: β-Caryophyllenic acid, β-caryophyllinic acid and β-caryophyllonic acid in different studies are the same tracer of β-caryophyllene-derived SOA (see its chemical structure in the Figure R2-2 below).

After derivatized with BSTFA, this compound has a molecular weight of 398 with fragment ions at m/z

383 and m/z 309 in the EI mass spectrum. Since there is a double-bond left, β-caryophyllenic acid is the accurate name.

[Figure]

Figure R2-2 The GC/MS EICs obtained for the trimethylsilylated extract of $PM_{2.5}$ aerosols and the chemical structure of β-caryophyllenic acid.

6. - Section 3.3: The impact of anthropogenic emissions to BSOA is perhaps one of the most important implications in this manuscript, but the current discussion appears weak. Based on the slopes obtained from the correlation studies, the authors implies that reducing $SO_4$ and $O_x$ in the atmosphere will lead to reduction of BSOA. However, this type of correlation analysis exhibits only correlation but not causation. The authors should justify why reducing anthropogenic emissions can likely reduce BSOA.

One way to do this, I think, is to add more discussion on the mechanisms behind the influence of anthropogenic emissions to BSOA.

Reply: Thanks for the suggestion. In the revised manuscript, we add more discussion on the mechanisms behind the influence of anthropogenic emissions to BSOA in Line 309-323 "It is interesting to note that $SOA_M$, $SOA_I$ and $SOA_C$ all positively correlated with sulfate and $O_x$ in the PRD

(Table 2). Since anthropogenic emissions can enhance BSOA formation (Hoyle et al., 2011), the reduction of anthropogenic emissions indeed lowers BSOA production (Carlton et al., 2018). As the oxidation product of $SO_2$, sulfate is a key species in particles that determines aerosol acidity and surface areas (Xu et al., 2015, 2016) which could promotes BSOA formation through the acid-catalyzed heterogeneous reactions. Recent study found that $SO_2$ could directly reaction with organic peroxides of monoterpene ozonolysis and form substantial organosulfates (Ye et al., 2018).

Thus, the decrease of $SO_2$ emission indeed reduces $SO_2$ and sulfate in the ambient air, which hereby leads to less acidic particles and reduces the BSOA production. For $O_x$, the increase of $O_3$ likely results in significant SOA formation through the BVOCs ozonolysis (Sipilä et al., 2014; Riva et al., 2017).

Hence, the decrease of $O_x$ resulting from the control of VOCs and $NO_x$ emissions could reduce BSOA

formation through $O_3$ chemistry. Based on the observed sulfate and $O_x$ dependence of BSOA in this study, the reduction of 1 μg m$^{-3}$ in sulfate and $O_x$ in the air of PRD could lower BSOA levels by 0.17

and 0.02 μg m$^{-3}$, respectively. If both concentrations decline by 50%, the reduction of $O_x$ is more efficient than sulfate in reducing BSOA in the PRD (Table 2)."

Reference
1. Carlton, A. G., Pye, H. O. T., Baker, K. R., and Hennigan, C. J.: Additional benefits of federal
air-quality rules: Model estimates of controllable biogenic secondary organic aerosol, Environ. Sci.
Technol., 52, 9254-9265, 10.1021/acs.est.8b01869, 2018.
2. Hoyle, C. R., Boy, M., Donahue, N. M., Fry, J. L., Glasius, M., Guenther, A., Hallar, A. G., Huff
Hartz, K., Petters, M. D., Petäjä, T., Rosenoern, T., and Sullivan, A. P.: A review of the anthropogenic
influence on biogenic secondary organic aerosol, Atmos. Chem. Phys., 11, 321-343,
10.5194/acp-11-321-2011, 2011.
3. Riva, M., Budisulistiorini, S. H., Zhang, Z., Gold, A., Thornton, J. A., Turpin, B. J., and Surratt, J. D.:
Multiphase reactivity of gaseous hydroperoxide oligomers produced from isoprene ozonolysis in the
presence of acidified aerosols, Atmos. Environ., 152, 314-322, 10.1016/j.atmosenv.2016.12.040, 2017.
4. Sipilä, M., Jokinen, T., Berndt, T., Richters, S., Makkonen, R., Donahue, N. M., Mauldin Iii, R. L.,
Kurtén, T., Paasonen, P., Sarnela, N., Ehn, M., Junninen, H., Rissanen, M. P., Thornton, J., Stratmann,
F., Herrmann, H., Worsnop, D. R., Kulmala, M., Kerminen, V. M., and Petäjä, T.: Reactivity of stabilized Criegee intermediates (sCIs) from isoprene and monoterpene ozonolysis toward $SO_2$ and organic acids, Atmos. Chem. Phys., 14, 12143-12153, 10.5194/acp-14-12143-2014, 2014.

5. Xu, L., Guo, H., Boyd, C. M., Klein, M., Bougiatioti, A., Cerully, K. M., Hite, J. R.,

Isaacman-VanWertz, G., Kreisberg, N. M., Knote, C., Olson, K., Koss, A., Goldstein, A. H., Hering, S.

V., de Gouw, J., Baumann, K., Lee, S.-H., Nenes, A., Weber, R. J., and Ng, N. L.: Effects of anthropogenic emissions on aerosol formation from isoprene and monoterpenes in the southeastern

United States, P. Natl. Acad. Sci. USA., 112, 37-42, 10.1073/ P.Natl.Acad.Sci.USA.1417609112, 2015.

6. Xu, L., Middlebrook, A. M., Liao, J., Gouw, J. A., Guo, H., Weber, R. J., Nenes, A., Lopez-Hilfiker,

F. D., Lee, B. H., Thornton, J. A., Brock, C. A., Neuman, J. A., Nowak, J. B., Pollack, I. B., Welti, A.,

Graus, M., Warneke, C., and Ng, N. L.: Enhanced formation of isoprene-derived organic aerosol in sulfur-rich power plant plumes during Southeast Nexus, J. Geophys. Res-Atmos., 121, 11,137-111,153, doi:10.1002/2016JD025156, 2016.

7. Ye, J., Abbatt, J. P. D., and Chan, A. W. H.: Novel pathway of $SO_2$ oxidation in the atmosphere:

Reactions with monoterpene ozonolysis intermediates and secondary organic aerosol, Atmos. Chem.

Phys., 18, 5549-5565, 10.5194/acp-18-5549-2018, 2018.

---

## Author Comment (AC3) · 8 Oct 2019

- 1 Impact of anthropogenic emissions on biogenic secondary
- 2 organic aerosol: Observation in the Pearl River Delta,

**3 South China**

4 Yu-Qing Zhang1, \*, Duo-Hong Chen2, \*, Xiang Ding1, †, Jun Li1, Tao Zhang2, Jun-Qi

5 Wang1, Qian Cheng1, Hao Jiang1, Wei Song1, Yu-Bo Ou2, Peng-Lin Ye3, Gan Zhang1,
6 Xin-Ming Wang1,4

- 6 Ani-Willig Wally
- 7 1 State Key Laboratory of Organic Geochemistry and Guangdong Provincial Key Laboratory of
- 8 Environmental Protection and Resources Utilization, Guangzhou Institute of Geochemistry, Chinese
- 9 Academy of Sciences, Guangzhou, 510640, China
- 10 2 State Environmental Protection Key Laboratory of Regional Air Quality Monitoring, Environmental
- 11 Monitoring Center of Guangdong Province, Guangzhou, 510308, China
- 12 3 Aerodyne Research Inc., Billerica, Massachusetts 01821, United States
- 13 4 Center for Excellence in Regional Atmospheric Environment, Institute of Urban Environment, Chinese

- 14 Academy of Sciences, Xiamen, 361021, China
- 15 \* These authors contributed equally to this work.
- 16 *† Correspondence to*: Xiang Ding (xiangd@gig.ac.cn)

Abstract. Secondary organic aerosol (SOA) formation from biogenic precursors is affected by 18 19 anthropogenic emissions, which is not well understood in polluted areas. In the study, we accomplished 20 a year-round campaign at nine sites in the polluted areas located in Pearl River Delta (PRD) region during 21 2015. We measured typical biogenic SOA (BSOA) tracers from isoprene, monoterpenes, and  $\beta$ -22 caryophyllene as well as major gaseous and particulate pollutants and investigated the impact of 23 anthropogenic pollutants on BSOA formation. The concentrations of BSOA tracers were in the range of 24 45.4 to 109 ng m-3 with the majority composed of products from monoterpenes (SOAM,  $47.2 \pm 9.29$  ng 25 m-3), followed by isoprene (SOAI, 23.1  $\pm$  10.8 ng m-3), and  $\beta$ -caryophyllene (SOAC, 3.85  $\pm$  1.75 ng m-3). 26 We found that atmospheric oxidants,  $O_x$  (O3 plus NO2), and sulfate correlated well with later-generation 27 SOAM tracers, but not so for first-generation SOAM products. This suggested that high Ox and sulfate 28 could promote the formation of later-generation SOAM products, which probably led to relatively aged 29 SOAM we observed in the PRD. For the SOAI tracers, not only 2-methylglyceric acid (NO/NO2-channel 30 product), but also the ratio of 2-methylglyceric acid to 2-methyltetrols (HO2-channel products) exhibit 31 NOx dependence, indicating the significant impact of NOx on SOAI formation pathways. The SOAC 
[revised manuscript text omitted]
 (2.35 ± 0.95  $\mu$ g m-3) and lowest in spring (1.06 ± 0.42  $\mu$ g m-3). SOAM contributions ranged from 57% 删除的内容: The oxidation of  $\beta$ -caryophyllene by the OH radical and O3 is very rapid with the lifetimes less than 10 min under typical conditions in the air of PRD (Table S5). The increase of sulfate could not only raise aerosol acidity but also enhance the salting-in effect (Xu et al., 2015). In the PRD, both Ox (Figure 3f) and sulfate (Figure S1) increased during winter, which could promote SOAC formation.

367 in winter to 68% in spring. The shares of  $SOA_1$  were only 5% in winter and reached up to 22% in summer.

368 The contributions of SOAC increased to 40% in wintertime.

369 It is interesting to note that SOAM, SOAI and SOAC all positively correlated with sulfate and Ox in 370 the PRD (Table 2). Since anthropogenic emissions can enhance BSOA formation (Hoyle et al., 2011), 371 the reduction of anthropogenic emissions indeed lowers BSOA production (Carlton et al., 2018). As the 372 oxidation product of SO2, sulfate is a key species in particles that determines aerosol acidity and surface 373 areas (Xu et al., 2015, 2016) which could promotes BSOA formation through the acid-catalyzed 374 heterogeneous reactions. Recent study found that SO2 could directly reaction with organic peroxides of 375 monoterpene ozonolysis and form substantial organosulfates (Ye et al., 2018). Thus, the decrease of SO2 376 emission indeed reduces SO2 and sulfate in the ambient air, which hereby leads to less acidic particles 377 and reduces the BSOA production. For Ox, the increase of O3 likely results in significant SOA formation 378 through the BVOCs ozonolysis (Sipilä et al., 2014; Riva et al., 2017). Hence, the decrease of Ox resulting 379 from the control of VOCs and NOx emissions could reduce BSOA formation through O3 chemistry. Based 380 on the observed sulfate and Ox 
[revised manuscript text omitted]

- 657

---

## Author Comment (AC4) · 8 Oct 2019

The comment was uploaded in the form of a supplement:

Please also note the supplement to this comment:
https://www.atmos-chem-phys-discuss.net/acp-2019-559/acp-2019-559-AC4-supplement.pdf
* * *

---

## Author Comment (AC5) · 8 Oct 2019

The comment was uploaded in the form of a supplement:

Please also note the supplement to this comment:
https://www.atmos-chem-phys-discuss.net/acp-2019-559/acp-2019-559-AC5-supplement.pdf
* * *

---

## Author Comment (AC6) · 8 Oct 2019

**Supporting Information**

**Impact of anthropogenic emissions on biogenic secondary organic aerosol: Observation in the Pearl River Delta, South China**

Yu-Qing Zhang[1, *], Duo-Hong Chen[2, *], Xiang Ding[1, †], Jun Li[1], Tao Zhang[2], Jun-Qi Wang[1], Qian Cheng[1, 3], Hao Jiang[1], Wei Song[1], Yu-Bo Ou[2], Peng-Lin Ye[3], Gan Zhang[1], Xin-Ming Wang[1, 4]

1 State Key Laboratory of Organic Geochemistry and Guangdong Provincial Key Laboratory of Environmental Protection and Resources Utilization, Guangzhou Institute of Geochemistry, Chinese Academy of Sciences, Guangzhou, 510640, China
2 State Environmental Protection Key Laboratory of Regional Air Quality Monitoring, Environmental Monitoring Center of Guangdong Province, Guangzhou, 510308, China
3 Aerodyne Research Inc., Billerica, Massachusetts 01821, United States
4 Center for Excellence in Regional Atmospheric Environment, Institute of Urban Environment, Chinese Academy of Sciences, Xiamen, 361021, China

* These authors contributed equally to this work.
† *Correspondence to*: Xiang Ding (xiangd@gig.ac.cn)

**Contents of this file**

Text S1
Figure S1 to S10
Table S1 to S7

**Text S1 SOA-tracer method for source apportionment**

The SOA-tracer method is developed by Kleindienst and co-workers. Based on chamber experiments, they determine the mass fractions of tracers in SOA ($f_{SOA}$) and SOC ($f_{SOC}$) for individual precursor:

$$f_{SOA} = \frac{\sum_i [\text{tr}_i]}{[SOA]} , \quad f_{SOC} = \frac{\sum_i [\text{tr}_i]}{[SOC]}$$

where $\sum_i [\text{tr}_i]$ is the sum of tracer concentrations for a precursor, and [SOA] and [SOC] are the measured SOA and SOC concentrations in chamber-generated SOA samples. The available $f_{SOA}$ and $f_{SOC}$ values were listed in Table S2. With these mass fractions in literatures and measured SOA tracers in the ambient air, SOA and SOC from different precursors have been estimated in different places of the world (Hu et al., 2008; Lewandowski et al., 2013; Stone et al., 2012; von Schneidemesser et al., 2009; Ding et al., 2014), with the assumption that the $f_{SOA}$ and $f_{SOC}$ values in the chamber samples are the same in the ambient air. In this study, the same set of SOA tracers reported by Kleindienst and co-workers were used for the SOC and SOA estimations (Table S2).

The uncertainty in the SOA-tracer method is induced from the analysis of organic tracers and the determination of conversion factors. The uncertainties in the tracers' analyses were estimated in the range of 15-157% (Table S2). The uncertainties in $f_{SOA}$ were reported to be 25% for isoprene, 48% for monoterpenes, and 22% for β-caryophyllene (Kleindienst et al., 2007; Lewandowski et al., 2013). Considering these factors, the uncertainty of the estimating procedure was calculated through error propagation. The relative standard deviations (RSD) were 37% for $SOA_I$, 67% for $SOA_M$, and 158% for $SOA_C$. On average, the RSD of total BSOA (sum of the three BVOCs) was 59%.

[Figure]

Figure S1 Seasonal variation of major components in PM$_{2.5}$. All the major components increased in winter and fall.

[Figure]

Figure S2 Spatial distribution of monoterpenes (a), and isoprene (b) emissions in the PRD (Zheng et al., 2010). The sampling sites are labeled.

[Figure]

Figure S3 Spatial distribution of (PNA+PA)/MBTCA ratios at 9 sites in the PRD. The (PNA+PA)/MBTCA ratios between two blue dash lines (1.51−5.91) indicate fresh SOA$_M$ from chamber studies (Eddingsaas et al., 2012; Offenberg et al., 2007). Box with error bars represent 10[th], 25[th], 75[th], 90[th] percentiles at each site. The line in the box is the median at each site. The data at WQS site during 2008 in the PRD (Ding et al., 2012) and at 12 sites during 2012-2013 in China (Ding et al., 2016) were reported in our previous studies.

[Figure]

Figure S4 Correlations of SOA$_{M\_F}$ tracers and SOA$_{M\_L}$ tracers with O$_x$

[Figure]

Figure S5 Correlations of SOA$_{M\_F}$ tracers and SOA$_{M\_L}$ tracers with sulfate

[Figure]

Figure S6 Correlations of SOA$_{HO2-channel}$ tracers and SOA$_{NO/NO2-channel}$ tracers with O$_x$ (a, b) and sulfate (c, d)

[Figure]

Figure S7 Significant correlations between 2-methyltetrol isomers at 9 sites in the PRD. K indicates the slope of each linear regression. The dash lines indicate the ratio range of 2-methyltetrol isomers in the SOA from isoprene ozonolysis (Riva et al., 2016).

[Figure]

Figure S8. Intercomparison of C₅-alkene triols compositions at 9 sites and in β-IEPOX and δ-IEPOX derived SOA (Lin et al., 2012).

[Figure]

Figure S9 Significant correlations among the SOA$_I$ tracers. K indicates the slope of each linear regression.

[Figure]

Figure S10 Long-term variation of sulfate in fall at WQS from 2007-2016.

Table S1 Data summary of gaseous and particulate species in the air of PRD

| | Zhaoqing (ZQ, urban site) | | | | | Guangzhou (GZ, urban site) | | | | | Dongguan (DG, urban site) | | | | | Nansha (NS, sub-urban site) | | | | | Zhuhai (ZH, sub-urban site) | | | | |
|---|---|---|---|---|---|---|---|---|---|---|---|---|---|---|---|---|---|---|---|---|---|---|---|---|---|
| | Winter | Spring | Summer | Fall | Annual | Winter | Spring | Summer | Fall | Annual | Winter | Spring | Summer | Fall | Annual | Winter | Spring | Summer | Fall | Annual | Winter | Spring | Summer | Fall | Annual |
| Temperature (°C) | 15.1 | 22.8 | 29.1 | 24.2 | 22.7(12.8-31.3) | 15.7 | 23.0 | 29.9 | 26.6 | 24.0(11.2-31.8) | 18.1 | 23.0 | 30.3 | 25.3 | 24.9(15.9-32.0) | 19.6 | 22.2 | 29.5 | 25.5 | 25.6(16.0-30.9) | 16.6 | 22.5 | 29.5 | 24.4 | 24.2(14.7-31.1) |
| RH (%) | 54 | 61 | 63 | 59 | 59(34-71) | 52 | 59 | 64 | 55 | 58(26-83) | 57 | 61 | 64 | 60 | 61(30-78) | 72 | 67 | 68 | 63 | 67(33-82) | 76 | 72 | 75 | 74 | 74(42-85) |
| SO$_2$ (µg m$^{-3}$) | 22.0 | 31.1 | 19.7 | 29.5 | 25.5(4.09-21.9) | 23.4 | 15.7 | 7.38 | 15.4 | 15.1(3.43-41.3) | 27.8 | 14.5 | 13.4 | 15.6 | 16.2(5.04-33.2) | 22.9 | 12.8 | 9.30 | 20.5 | 14.4(4.04-36.5) | 8.55 | 6.05 | 5.77 | 10.4 | 7.33(2.14-14.6) |
| NO$_2$ (µg m$^{-3}$) | 40.4 | 24.1 | 15.4 | 36.7 | 29.1(2.45-82.7) | 84.8 | 49.3 | 36.3 | 61.2 | 57.2(29.7-155) | 73.7 | 34.0 | 24.2 | 35.4 | 36.9(10.5-102) | 61.4 | 27.6 | 19.0 | 44.4 | 31.4(8.08-91.7) | 50.7 | 21.7 | 14.3 | 34.2 | 26.8(7.08-65.0) |
| NO (µg m$^{-3}$) | 14.7 | 4.65 | 4.06 | 5.88 | 7.31(2.00-35.4) | 31.8 | 6.73 | 5.24 | 6.65 | 12.7(1.13-126) | 25.1 | 3.64 | 6.77 | 6.14 | 7.88(1.5-42.4) | 13.5 | 2.48 | 2.00 | 6.16 | 4.04(0.56-13.5) | 6.75 | na | na | na | 6.75(5.63-7.53) |
| NO$_x$ (µg m$^{-3}$) | 62.9 | 31.3 | 21.5 | 45.7 | 40.3(6.70-121) | 134 | 59.5 | 44.3 | 71.4 | 76.8(35.2-349) | 112 | 39.6 | 34.6 | 44.8 | 49.0(13.6-167) | 83.0 | 32.1 | 22.7 | 54.4 | 38.3(10.3-111) | 57.5 | na | na | na | 57.4(47.6-72.1) |
| O$_3$ (µg m$^{-3}$) | 55.2 | 64.3 | 81.9 | 59.6 | 65.2(11.8-145) | 52.9 | 54.1 | 51.2 | 62.2 | 54.7(18.8-115) | 42.3 | 65.6 | 80.0 | 66.9 | 66.6(31.5-123) | 64.8 | 80.2 | 80.8 | 78.6 | 79.0(21.3-149) | 49.7 | 48.7 | 66.3 | 106 | 67.5(18.3-155) |
| O$_x$ (µg m$^{-3}$) | 95.7 | 88.5 | 97.2 | 96.3 | 94.4(49.9-173) | 138 | 103 | 87.5 | 123 | 112(55.3-208) | 116 | 99.7 | 104 | 102 | 103(46.0-160) | 126 | 108 | 99.8 | 123 | 110(47.7-198) | 100 | 70.4 | 80.6 | 140 | 94.3(35.7-190) |
| CO (mg m$^{-3}$) | 1.18 | 0.69 | 0.66 | 0.73 | 0.81(0.21-1.66) | 1.21 | 1.04 | 0.73 | 0.78 | 0.94(0.52-1.81) | 1.15 | 0.72 | 0.71 | 0.61 | 0.73(0.32-1.52) | 0.85 | 0.74 | 0.66 | 0.69 | 0.70(0.37-1.21) | 1.11 | 0.66 | 0.62 | 0.59 | 0.70(0.48-1.14) |
| OC (µgC m$^{-3}$) | 21.5 | 8.26 | 8.73 | 9.60 | 12.0(4.66-32.1) | 15.9 | 6.55 | 5.90 | 10.2 | 9.59(3.12-33.5) | 20.2 | 6.51 | 6.28 | 7.09 | 8.34(3.46-27.7) | 17.1 | 5.37 | 5.43 | 9.18 | 7.20(1.94-19.6) | 9.82 | 4.41 | 4.50 | 8.55 | 6.05(1.94-17.5) |
| EC (µgC m$^{-3}$) | 5.70 | 1.50 | 1.80 | 2.34 | 2.83(0.79-8.41) | 4.35 | 1.58 | 1.60 | 2.51 | 2.51(0.79-11.7) | 6.76 | 1.80 | 2.12 | 3.39 | 2.99(0.84-8.39) | 5.32 | 1.22 | 1.48 | 2.71 | 1.99(0.44-6.61) | 2.82 | 1.07 | 0.96 | 2.55 | 1.59(0.44-5.16) |
| SO$_4^{2-}$ (µg m$^{-3}$) | 12.8 | 8.27 | 8.86 | 10.1 | 10.0(2.92-20.9) | 12.0 | 6.72 | 5.78 | 9.55 | 8.44(2.61-19.5) | 17.7 | 6.87 | 6.52 | 8.09 | 8.52(2.45-21.2) | 14.1 | 7.48 | 6.18 | 10.6 | 8.32(2.18-16.9) | 14.0 | 6.80 | 5.48 | 12.3 | 8.47(2.33-21.9) |
| NO$_3^-$ (µg m$^{-3}$) | 10.8 | 4.05 | 0.54 | 1.31 | 4.18(0.14-21.5) | 9.11 | 1.82 | 0.57 | 1.29 | 3.22(0.15-23.3) | 12.3 | 2.92 | 0.75 | 0.79 | 2.88(0.23-16.2) | 14.4 | 1.59 | 0.50 | 1.06 | 1.81(0.23-14.4) | 4.97 | 1.28 | 0.16 | 1.53 | 1.38(0.04-5.91) |
| NH$_4^+$ (µg m$^{-3}$) | 7.92 | 4.44 | 3.70 | 3.86 | 4.98(1.10-14.3) | 7.10 | 2.98 | 2.26 | 3.87 | 4.03(0.95-12.8) | 8.58 | 3.03 | 2.09 | 2.79 | 3.41(0.62-9.75) | 11.5 | 3.54 | 2.33 | 3.91 | 3.69(0.93-11.5) | 6.35 | 2.51 | 2.23 | 4.54 | 3.34(0.60-8.10) |
| Cl$^-$ (µg m$^{-3}$) | 1.42 | 0.50 | 0.08 | 0.19 | 0.55(0.03-2.89) | 0.86 | 0.37 | 0.13 | 0.14 | 0.37(0.06-1.78) | 1.84 | 0.24 | 0.14 | 0.06 | 0.36(0.03-2.21) | 1.33 | 0.44 | 0.09 | 0.20 | 0.30(0.04-1.44) | 0.27 | 0.25 | 0.05 | 0.06 | 0.14(0.01-0.52) |
| Na$^+$ (µg m$^{-3}$) | 0.83 | 0.34 | 0.29 | 0.36 | 0.45(0.08-2.66) | 0.66 | 0.23 | 0.26 | 0.40 | 0.39(0.08-1.13) | 0.73 | 0.38 | 0.51 | 0.38 | 0.46(0.11-0.96) | 0.60 | 1.46 | 0.27 | 0.43 | 0.68(0.15-2.30) | 0.48 | 0.46 | 0.28 | 0.34 | 0.37(0.09-0.71) |
| K$^+$ (µg m$^{-3}$) | 0.83 | 0.38 | 0.23 | 0.33 | 0.44(0.11-1.32) | 0.70 | 0.21 | 0.25 | 0.36 | 0.38(0.01-2.16) | 1.45 | 0.36 | 0.20 | 0.35 | 0.45(0.14-1.95) | 0.81 | 0.30 | 0.11 | 0.30 | 0.26(0.04-0.81) | 0.46 | 0.22 | 0.09 | 0.24 | 0.21(0.02-0.58) |
| Mg$^{2+}$ (µg m$^{-3}$) | 0.10 | 0.05 | 0.04 | 0.04 | 0.06(0.02-0.35) | 0.08 | 0.04 | 0.03 | 0.04 | 0.05(0.01-0.18) | 0.08 | 0.06 | 0.07 | 0.04 | 0.06(0.02-0.12) | 0.07 | 0.15 | 0.11 | 0.08 | 0.11(0.03-0.21) | 0.06 | 0.09 | 0.04 | 0.04 | 0.06(0.01-0.13) |
| Ca$^{2+}$ (µg m$^{-3}$) | 0.59 | 0.31 | 0.37 | 0.44 | 0.44(0.07-1.14) | 0.49 | 0.34 | 0.24 | 0.30 | 0.34(0.17-0.88) | 0.42 | 0.32 | 0.32 | 0.30 | 0.33(0.11-0.79) | 0.32 | 0.46 | 0.41 | 0.44 | 0.43(0.17-1.03) | 0.22 | 0.29 | 0.07 | 0.17 | 0.18(0.01-0.85) |
| PM$_{2.5}$ (µg m$^{-3}$) | 60.7 | 25.4 | 27.3 | 36.9 | 37.5(11.7-85.6) | 64.5 | 31.4 | 20.7 | 35.1 | 37.6(10.1-131) | 102 | 39.5 | 26.0 | 32.9 | 41.9(14.7-125) | 78.1 | 24.2 | 24.6 | 36.8 | 31.2(7.74-78.9) | 51.8 | 27.1 | 17.4 | 43.9 | 30.5(9.46-83.3) |
| | | | | | | | | | | | | | | | | | | | | | | | | | |
| 3-Hydroxyglutaric acid | 21.5 | 18.0 | 35.9 | 20.1 | 23.8(3.32-89.5) | 18.5 | 13.4 | 19.9 | 32.7 | 20.9(2.77-54.0) | 36.5 | 17.6 | 27.6 | 20.2 | 23.2(3.73-73.6) | 28.0 | 7.80 | 11.0 | 9.26 | 10.5(0.62-27.9) | 16.4 | 6.13 | 15.0 | 33.7 | 16.9(0.70-61.7) |
| 3-Hydroxy-4,4-dimethylglutaric acid | 10.6 | 12.3 | 24.2 | 15.5 | 15.6(1.51-57.4) | 8.09 | 8.39 | 14.8 | 24.4 | 13.7(nd-35.8) | 19.3 | 12.8 | 23.9 | 18.1 | 18.0(1.17-60.5) | 12.6 | 6.39 | 10.4 | 5.60 | 7.93(0.27-28.5) | 14.8 | 6.22 | 12.7 | 30.5 | 15.1(0.56-53.4) |
| cis-Pinonic acid | 5.04 | 3.76 | 2.77 | 4.37 | 3.98(0.30-10.5) | 6.11 | 5.67 | 2.19 | 6.77 | 4.99(0.36-20.3) | 1.65 | 3.26 | 0.39 | 1.66 | 1.84(nd-11.8) | 0.57 | 3.48 | 0.97 | 1.54 | 1.85(0.06-12.8) | 0.37 | 3.01 | 0.83 | 2.02 | 1.62(0.14-10.3) |
| Pinic acid | 1.40 | 1.03 | 1.40 | 2.33 | 1.54(0.10-5.10) | 0.99 | 0.76 | 0.57 | 1.48 | 0.92(0.11-3.47) | 0.87 | 0.94 | 0.46 | 0.76 | 0.75(0.12-2.73) | 0.51 | 0.77 | 0.52 | 0.71 | 0.65(0.05-2.69) | 0.37 | 0.31 | 0.52 | 0.63 | 0.45(nd-1.82) |
| 3-Methyl-1,2,3-butanetricarboxylic acid | 6.65 | 4.43 | 17.8 | 8.47 | 9.32(0.90-55.5) | 7.70 | 5.69 | 10.0 | 14.5 | 9.44(0.55-25.3) | 6.67 | 3.05 | 7.96 | 10.3 | 6.99(0.17-23.5) | 6.76 | 2.34 | 6.90 | 6.80 | 5.52(0.07-21.0) | 2.87 | 1.42 | 5.44 | 15.4 | 6.11(0.17-35.5) |
| Sum of SOA$_M$ tracers | 45.1 | 39.4 | 82.0 | 50.8 | 54.3(10.0-205) | 41.4 | 33.9 | 47.4 | 79.9 | 50.0(9.79-118) | 65.0 | 37.7 | 60.3 | 51.0 | 50.9(8.57-156) | 48.5 | 20.8 | 29.8 | 23.9 | 26.5(3.24-67.3) | 34.7 | 17.1 | 34.5 | 82.3 | 40.3(1.39-153) |
| | | | | | | | | | | | | | | | | | | | | | | | | | |
| cis-3-Methyltetrahydrofuran-3,4-diol | 0.02 | 0.03 | 0.13 | 0.09 | 0.06(nd-0.40) | 0.02 | 0.04 | 0.06 | 0.10 | 0.06(nd-0.14) | 0.02 | 0.02 | 0.04 | 0.09 | 0.04(nd-0.26) | nd | nd | 0.07 | 0.11 | 0.09(0.02-0.14) | nd | nd | 0.01 | 0.02 | 0.01(nd-0.05) |
| trans-3-Methyltetrahydrofuran-3,4-diol | 0.04 | 0.08 | 0.64 | 0.25 | 0.25(nd-2.06) | 0.04 | 0.05 | 0.16 | 0.21 | 0.12(nd-0.39) | 0.05 | 0.04 | 0.10 | 0.22 | 0.11(nd-0.67) | nd | nd | 0.13 | 0.31 | 0.21(0.01-0.67) | 0.01 | 0.02 | 0.06 | 0.08 | 0.04(nd-0.18) |
| cis-2-Methyl-1,3,4-trihydroxy-1-butene | 0.59 | 1.63 | 11.3 | 6.26 | 4.93(0.11-33.2) | 0.39 | 0.85 | 3.32 | 4.50 | 2.27(0.02-10.6) | 0.49 | 0.28 | 0.76 | 0.37 | 0.45(0.02-1.54) | 0.43 | 0.23 | 1.72 | 2.50 | 1.37(0.05-5.25) | 0.12 | 0.26 | 1.36 | 1.85 | 0.95(0.04-4.24) |
| 3-Methyl-2,3,4-trihydroxy-1-butene | 0.21 | 0.82 | 5.82 | 3.19 | 2.50(0.07-17.0) | 0.24 | 0.46 | 1.85 | 2.49 | 1.26(0.01-5.85) | 0.20 | 0.21 | 0.68 | 0.70 | 0.46(0.01-2.52) | 0.33 | 0.12 | 1.01 | 1.65 | 0.85(nd-3.19) | 0.05 | 0.13 | 0.75 | 0.89 | 0.46(0.01-2.31) |
| trans-2-Methyl-1,3,4-trihydroxy-1-butene | 1.62 | 4.71 | 25.9 | 13.5 | 11.4(0.17-80.4) | 0.95 | 1.86 | 7.49 | 11.5 | 5.43(0.11-29.2) | 1.21 | 0.58 | 1.59 | 1.06 | 1.06(0.01-3.90) | 1.24 | 0.64 | 3.46 | 5.11 | 2.85(0.09-11.3) | 4.97 | 0.53 | 2.27 | 3.91 | 1.81(0.02-9.74) |
| 2-Methylglyceric acid | 2.02 | 3.31 | 3.00 | 3.84 | 3.04(0.24-9.02) | 1.32 | 1.04 | 0.79 | 4.02 | 1.70(0.09-10.3) | 1.56 | 1.41 | 0.94 | 3.09 | 1.83(0.07-7.75) | 1.25 | 0.73 | 1.33 | 2.30 | 1.43(0.09-7.43) | 0.47 | 0.37 | 0.78 | 1.26 | 0.71(0.09-3.11) |
| 2-Methylthreitol | 1.36 | 7.08 | 16.7 | 6.56 | 7.93(0.68-33.9) | 0.91 | 1.65 | 4.92 | 5.23 | 3.22(0.20-15.0) | 1.45 | 2.08 | 3.16 | 6.34 | 3.60(0.22-17.3) | 0.97 | 0.54 | 4.08 | 4.17 | 2.88(0.11-11.9) | 0.20 | 0.82 | 3.38 | 2.48 | 1.93(0.09-12.0) |
| 2-Methylerythritol | 3.31 | 15.5 | 42.4 | 15.5 | 19.1(1.59-74.7) | 1.94 | 3.87 | 15.7 | 11.6 | 8.58(0.41-49.3) | 2.97 | 4.56 | 8.67 | 14.7 | 8.56(0.48-42.6) | 1.77 | 1.23 | 12.5 | 9.83 | 7.77(0.27-37.6) | 0.44 | 1.86 | 9.49 | 4.69 | 4.79(0.25-33.9) |
| Sum of SOA$_I$ tracers | 9.18 | 33.1 | 106 | 49.2 | 49.3(4.86-250) | 5.80 | 9.77 | 34.2 | 39.6 | 22.6(0.89-97.9) | 7.95 | 9.18 | 15.9 | 26.2 | 16.0(0.95-68.6) | 6.00 | 3.49 | 24.2 | 24.0 | 17.0(0.83-69.9) | 1.53 | 3.99 | 18.1 | 15.2 | 10.8(0.54-62.2) |
| | | | | | | | | | | | | | | | | | | | | | | | | | |
| β-Caryophyllenic acid | 9.76 | 3.70 | 4.02 | 3.40 | 5.22(0.40-14.3) | 11.4 | 4.33 | 4.75 | 7.78 | 7.07(nd-20.5) | 9.05 | 2.32 | 2.35 | 2.22 | 3.13(0.07-10.8) | 8.40 | 1.81 | 1.20 | 1.47 | 1.88(0.20-8.40) | 3.69 | 0.67 | 1.03 | 3.05 | 1.82(nd-5.53) |
| | | | | | | | | | | | | | | | | | | | | | | | | | |
| SOA$_M$ (µg m$^{-3}$) | 1.02 | 0.90 | 1.86 | 1.15 | 1.23(0.22-4.65) | 0.94 | 0.77 | 1.08 | 1.81 | 1.13(0.22-2.69) | 1.48 | 0.86 | 1.37 | 1.16 | 1.15(0.19-3.55) | 1.10 | 0.47 | 0.68 | 0.54 | 0.60(0.07-1.52) | 0.79 | 0.39 | 0.78 | 1.87 | 0.91(0.04-3.48) |
| SOA$_I$ (µg m$^{-3}$) | 0.11 | 0.41 | 0.99 | 0.41 | 0.47(0.05-1.85) | 0.07 | 0.10 | 0.34 | 0.33 | 0.21(0.01-1.04) | 0.09 | 0.13 | 0.20 | 0.38 | 0.22(0.01-1.07) | 0.06 | 0.04 | 0.28 | 0.26 | 0.19(0.01-0.83) | 0.02 | 0.05 | 0.22 | 0.13 | 0.11(0.01-0.76) |
| SOA$_C$ (µg m$^{-3}$) | 0.90 | 0.34 | 0.37 | 0.31 | 0.47(0.03-1.31) | 1.05 | 0.40 | 0.44 | 0.71 | 0.64(nd-1.88) | 0.83 | 0.21 | 0.22 | 0.20 | 0.28(0.01-0.99) | 0.77 | 0.17 | 0.11 | 0.14 | 0.17(0.01-0.77) | 0.34 | 0.06 | 0.09 | 0.28 | 0.16(nd-0.50) |
| BSOA (µg m$^{-3}$) | 2.03 | 1.65 | 3.22 | 1.88 | 2.19(0.45-7.40) | 2.06 | 1.27 | 1.85 | 2.86 | 1.99(0.45-4.21) | 2.40 | 1.20 | 1.79 | 1.75 | 1.66(0.26-4.47) | 1.93 | 0.68 | 1.07 | 0.94 | 0.96(0.12-2.24) | 1.14 | 0.50 | 1.09 | 2.28 | 1.20(0.08-4.25) |

"na" means not available and "nd" means not detected.

Table S1 Data summary of gaseous and particulate species in the air of PRD (continued)

| | Tianhu (TH, rural site) | | | | | Boluo (BL, rural site) | | | | | Heshan (HS, rural site) | | | | | Taishan (TS, rural site) | | | | | 9 sites average | | | | |
|---|---|---|---|---|---|---|---|---|---|---|---|---|---|---|---|---|---|---|---|---|---|---|---|---|---|
| | Winter | Spring | Summer | Fall | Annual | Winter | Spring | Summer | Fall | Annual | Winter | Spring | Summer | Fall | Annual | Winter | Spring | Summer | Fall | Annual | Winter | Spring | Summer | Fall | Annual |
| Temperature (°C) | 13.2 | 19.9 | 27.0 | 24.4 | 20.5(11.0-29.4) | 16.4 | 20.5 | 28.6 | 23.4 | 22.7(13.9-31.4) | 13.1 | 21.8 | 29.0 | 23.1 | 21.4(10.5-31.0) | 16.4 | 23.0 | 29.1 | 23.4 | 22.9(14.0-31.1) | 16.0 | 22.1 | 29.1 | 24.5 | 23.2(10.5-32.0) |
| RH (%) | na | na | na | na | na | 75 | 75 | 70 | 71 | 72(60-85) | 58 | 64 | 70 | 63 | 63(39-86) | 75 | 76 | 71 | 75 | 74(54-84) | 58 | 60 | 61 | 58 | 59(26 -86) |
| $SO_2$ (µg m$^{-3}$) | 11.9 | 9.75 | 8.70 | 13.5 | 10.5(5.34-16.9) | 15.1 | 10.1 | 10.6 | 15.6 | 13.0(5.13-20.3) | 31.3 | 23.2 | 11.7 | 29.5 | 24.4(5.60-46.8) | 11.6 | 4.35 | 5.46 | 7.15 | 7.14(0.95-17.0) | 19.4 | 14.2 | 10.2 | 17.5 | 14.9(7.14-25.5) |
| $NO_2$ (µg m$^{-3}$) | 10.6 | 8.54 | 10.8 | 3.37 | 8.98(2.95-16.2) | 14.1 | 17.6 | 11.4 | 10.7 | 12.9(3.78-21.4) | 45.7 | 26.7 | 5.85 | 43.4 | 31.4(3.68-60.2) | 38.7 | 14.3 | 13.9 | 18.5 | 21.3(6.47-49.4) | 46.7 | 24.9 | 16.8 | 32.0 | 28.5(8.98-57.2) |
| $NO$ (µg m$^{-3}$) | 0.25 | 0.32 | 1.72 | 0.70 | 0.87(0.13-2.47) | 1.47 | 1.30 | 0.75 | 0.83 | 1.03(0.27-2.82) | 3.49 | 3.77 | 1.40 | 3.67 | 3.15(0.68-12.4) | 3.58 | 0.47 | 0.63 | 1.13 | 1.45(0.08-5.78) | 11.2 | 2.60 | 2.51 | 3.46 | 5.03(0.87-12.7) |
| $NO_x$ (µg m$^{-3}$) | 11.8 | 9.82 | 14.2 | 5.08 | 10.8(4.68-19.8) | 17.2 | 20.4 | 14.7 | 12.8 | 15.7(8.65-26.6) | 51.6 | 33.0 | 17.37 | 48.7 | 38.5(9.72-72.4) | 44.0 | 16.1 | 15.8 | 19.8 | 23.9(8.08-57.0) | 63.8 | 30.0 | 22.4 | 38.5 | 39.0(10.8-76.8) |
| $O_3$ (µg m$^{-3}$) | 99.2 | 84.6 | 90.2 | 139 | 97.2(52.8-150) | 29.6 | 38.0 | 69.8 | 55.8 | 50.6(18.2-97.3) | 48.7 | 75.4 | 61.0 | 60.4 | 61.3(12.8-135) | 32.9 | 65.6 | 77.4 | 87.9 | 65.9(10.9-147) | 52.8 | 64.1 | 73.2 | 79.6 | 67.7(50.6-97.2) |
| $O_x$ (µg m$^{-3}$) | 110 | 93.2 | 101 | 143 | 106(69.0-154) | 43.7 | 55.6 | 81.2 | 66.5 | 63.5(27.8-112) | 94.4 | 102 | 66.9 | 104 | 92.8(33.3-184) | 71.6 | 80.0 | 91.2 | 106 | 87.2(25.8-173) | 99.5 | 88.9 | 90.0 | 112 | 96.1(63.5-112) |
| CO (mg m$^{-3}$) | 0.62 | 0.57 | 0.56 | 0.32 | 0.54(0.26-0.87) | 0.85 | 0.67 | 0.51 | 0.54 | 0.62(0.30-1.06) | 1.10 | 0.88 | 0.87 | 0.82 | 0.91(0.50-1.22) | 0.88 | 0.56 | 0.54 | 0.78 | 0.68(0.54-0.94) | 1.00 | 0.73 | 0.65 | 0.65 | 0.74(0.54-0.94) |
| OC (µgC m$^{-3}$) | 8.05 | 5.38 | 6.30 | 8.05 | 6.49(3.64-10.4) | 7.40 | 6.63 | 7.03 | 8.67 | 7.52(2.64-16.7) | 12.9 | 6.82 | 6.31 | 12.6 | 9.65(2.74-22.4) | 10.8 | 6.79 | 8.28 | 12.8 | 9.67(4.24-23.1) | 13.7 | 6.30 | 6.53 | 8.74 | 8.50(1.93-33.4) |
| EC (µgC m$^{-3}$) | 1.44 | 0.96 | 1.04 | na | 1.13(0.40-2.04) | 2.98 | 1.99 | 1.92 | 2.08 | 2.22(0.54-8.22) | 3.38 | 1.86 | 1.36 | 3.51 | 2.52(0.52-6.19) | 3.22 | 1.28 | 1.66 | 3.14 | 2.32(0.64-6.86) | 4.00 | 1.47 | 1.55 | 2.47 | 2.23(0.40-11.6) |
| $SO_4^{2-}$ (µg m$^{-3}$) | 9.83 | 6.32 | 5.11 | na | 7.18(2.99-15.1) | 4.38 | 10.11 | 6.40 | 5.97 | 6.45(2.10-18.5) | 10.4 | 6.96 | 5.43 | 13.9 | 9.17(2.24-28.2) | 11.3 | 5.98 | 7.09 | 13.3 | 9.41(2.12-18.9) | 11.8 | 7.28 | 6.32 | 9.31 | 8.44(2.10-28.2) |
| $NO_3^-$ (µg m$^{-3}$) | 1.00 | 1.27 | 0.09 | na | 0.88(0.01-2.93) | 2.69 | 3.50 | 0.48 | 0.60 | 1.56(0.11-8.66) | 8.33 | 3.01 | 0.30 | 9.49 | 4.23(0.15-16.3) | 9.49 | 2.23 | 0.40 | 2.81 | 3.73(0.12-23.7) | 8.12 | 2.41 | 0.42 | 1.64 | 2.65(0.01-23.7) |
| $NH_4^+$ (µg m$^{-3}$) | 3.80 | 2.52 | 1.94 | na | 2.80(1.11-5.88) | 2.43 | 4.58 | 2.42 | 2.38 | 2.79(0.77-8.96) | 5.92 | 3.50 | 2.14 | 6.06 | 4.40(0.85-13.8) | 6.32 | 3.01 | 2.57 | 5.22 | 4.28(0.44-11.3) | 6.66 | 3.34 | 2.41 | 3.63 | 3.74(0.44-14.2) |
| $Cl^-$ (µg m$^{-3}$) | 0.05 | 0.07 | 0.02 | na | 0.04(nd-0.15) | 0.54 | 0.40 | 0.05 | 0.08 | 0.23(0.01-1.77) | 1.20 | 0.46 | 0.05 | 0.32 | 0.50(0.01-1.85) | 1.54 | 0.26 | 0.06 | 0.24 | 0.52(0.01-4.20) | 1.00 | 0.33 | 0.07 | 0.14 | 0.33(nd-4.20) |
| $Na^+$ (µg m$^{-3}$) | 0.57 | 0.48 | 0.14 | na | 0.42(0.04-0.75) | 0.10 | 0.26 | 0.43 | 0.25 | 0.27(0.03-1.00) | 0.71 | 0.31 | 0.24 | 0.59 | 0.46(0.18-1.29) | 0.49 | 0.28 | 0.35 | 0.70 | 0.45(0.18-0.96) | 0.57 | 0.47 | 0.31 | 0.38 | 0.44(0.03-2.66) |
| $K^+$ (µg m$^{-3}$) | 0.43 | 0.24 | 0.12 | na | 0.27(0.06-0.57) | 0.27 | 0.24 | 0.19 | 0.20 | 0.22(0.07-0.71) | 1.02 | 0.35 | 0.32 | 0.81 | 0.62(0.11-1.50) | 0.49 | 0.16 | 0.42 | 0.62 | 0.39(0.09-1.02) | 0.72 | 0.27 | 0.21 | 0.34 | 0.36(0.01-2.16) |
| $Mg^{2+}$ (µg m$^{-3}$) | 0.08 | 0.07 | 0.02 | na | 0.05(nd-0.10) | 0.02 | 0.03 | 0.03 | 0.04 | 0.03(0.01-0.07) | 0.04 | 0.03 | 0.02 | 0.04 | 0.03(nd-0.09) | 0.02 | 0.03 | 0.03 | 0.04 | 0.02(nd-0.06) | 0.06 | 0.06 | 0.04 | 0.04 | 0.05(nd-0.34) |
| $Ca^{2+}$ (µg m$^{-3}$) | 0.46 | 0.42 | 0.10 | na | 0.35(0.04-0.85) | 0.18 | 0.16 | 0.25 | 0.34 | 0.24(0.10-0.62) | 0.34 | 0.13 | 0.13 | 0.36 | 0.24(0.02-0.66) | 0.20 | 0.19 | 0.19 | 0.31 | 0.22(0.12-0.40) | 0.36 | 0.29 | 0.23 | 0.29 | 0.30(nd-1.14) |
| $PM_{2.5}$ (µg m$^{-3}$) | 33.9 | 21.6 | 18.9 | na | 25.0(9.98-43.0) | 31.4 | 30.5 | 24.5 | 28.7 | 28.4(11.0-72.5) | 63.1 | 29.8 | 21.4 | 54.9 | 42.2(6.78-112) | 55.3 | 21.1 | 24.5 | 52.6 | 38.3(7.68-114) | 60.1 | 27.8 | 22.8 | 35.7 | 34.7(6.78-131) |
| 3-Hydroxyglutaric acid | 21.0 | 18.1 | 22.0 | 35.6 | 22.2(4.44-52.4) | 14.1 | 16.4 | 24.3 | 16.5 | 18.1(3.64-74.8) | 16.3 | 14.7 | 24.0 | 35.7 | 22.6(2.72-89.4) | 19.0 | 5.25 | 23.4 | 42.0 | 22.4(1.44-79.2) | 21.2 | 13.1 | 22.6 | 27.3 | 20.1(10.5-23.8) |
| 3-Hydroxy-4,4-dimethylglutaric acid | 10.7 | 15.2 | 15.0 | 35.5 | 16.6(1.65-36.8) | 6.20 | 13.9 | 18.4 | 19.1 | 14.9(0.77-53.6) | 7.87 | 12.5 | 19.6 | 26.7 | 16.6(2.11-61.0) | 7.49 | 4.36 | 15.5 | 25.7 | 13.2(nd-47.9) | 10.8 | 10.2 | 17.2 | 22.3 | 14.7(7.93-18.0) |
| cis-Pinonic acid | 11.8 | 6.26 | 4.03 | 29.3 | 10.2(0.66-34.3) | 1.80 | 1.16 | 1.57 | 2.22 | 1.74(0.28-5.34) | 6.81 | 3.32 | 2.01 | 2.41 | 3.63(0.56-18.5) | 7.55 | 3.91 | 2.01 | 1.82 | 3.82(0.08-26.1) | 4.64 | 3.76 | 1.86 | 5.79 | 3.75(1.62-10.2) |
| Pinic acid | 1.70 | 0.92 | 1.33 | 6.58 | 1.99(0.19-7.69) | 0.73 | 0.49 | 0.66 | 0.94 | 0.72(0.12-1.80) | 1.17 | 0.65 | 0.77 | 1.45 | 1.01(0.14-3.17) | 0.63 | 0.95 | 1.02 | 1.44 | 1.00(0.13-2.43) | 0.93 | 0.76 | 0.81 | 1.81 | 1.01(0.45-1.99) |
| 3-Methyl-1,2,3-butanetricarboxylic acid | 4.48 | 3.76 | 5.37 | 18.3 | 6.31(0.75-18.8) | 6.89 | 3.68 | 12.1 | 10.4 | 8.91(0.35-34.5) | 3.18 | 7.44 | 14.1 | 9.73 | 8.62(0.92-18.1) | 6.98 | 2.08 | 6.68 | 13.8 | 7.39(0.40-25.8) | 5.80 | 3.76 | 9.60 | 12.0 | 7.63(5.52-9.44) |
| Sum of $SOA_M$ tracers | 49.7 | 44.3 | 47.7 | 125 | 57.4(15.5-134) | 29.7 | 35.6 | 57.0 | 49.2 | 44.5(7.51-164) | 35.3 | 38.6 | 60.5 | 76.0 | 52.5(12.3-167) | 41.7 | 16.5 | 48.6 | 84.9 | 47.9(4.82-157) | 43.5 | 31.6 | 52.0 | 69.2 | 47.1(26.5-57.4) |
| cis-3-Methyltetrahydrofuran-3,4-diol | 0.01 | 0.02 | 0.04 | 0.13 | 0.03(nd-0.16) | 0.02 | 0.02 | 0.07 | 0.14 | 0.07(nd-0.51) | 0.02 | 0.01 | 0.05 | 0.10 | 0.04(nd-0.22) | 0.02 | 0.06 | 0.05 | 0.12 | 0.06(nd-0.18) | 0.02 | 0.03 | 0.06 | 0.10 | 0.05(0.01-0.09) |
| trans-3-Methyltetrahydrofuran-3,4-diol | 0.02 | 0.04 | 0.08 | 0.33 | 0.08(nd-0.42) | 0.03 | 0.04 | 0.22 | 0.35 | 0.18(nd-1.21) | 0.03 | 0.03 | 0.24 | 0.30 | 0.15(nd-0.67) | 0.04 | 0.13 | 0.13 | 0.25 | 0.13(0.01-0.48) | 0.03 | 0.05 | 0.20 | 0.26 | 0.14(0.04-0.25) |
| cis-2-Methyl-1,3,4-trihydroxy-1-butene | 0.28 | 0.31 | 1.32 | 7.20 | 1.48(0.05-8.89) | 0.19 | 0.10 | 1.71 | 1.55 | 1.00(0.01-5.64) | 0.31 | 0.27 | 1.66 | 2.91 | 1.28(0.04-7.09) | 0.24 | 0.12 | 2.01 | 5.88 | 2.19(0.03-12.0) | 0.34 | 0.45 | 2.80 | 3.67 | 1.77(0.45-4.93) |
| 3-Methyl-2,3,4-trihydroxy-1-butene | 0.17 | 0.19 | 0.89 | 5.49 | 1.07(nd-6.86) | 0.13 | 0.08 | 1.41 | 1.18 | 0.78(nd-4.53) | 0.16 | 0.22 | 1.53 | 2.38 | 1.07(nd-5.23) | 0.14 | 0.04 | 1.06 | 2.82 | 1.15(nd-5.94) | 0.18 | 0.25 | 1.67 | 2.31 | 1.07(0.46-2.50) |
| trans-2-Methyl-1,3,4-trihydroxy-1-butene | 0.67 | 0.71 | 2.98 | 15.6 | 3.29(0.15-17.8) | 0.44 | 0.30 | 3.70 | 3.17 | 2.11(0.04-12.3) | 0.77 | 0.71 | 3.70 | 7.29 | 3.11(0.10-17.1) | 0.56 | 0.16 | 4.28 | 13.3 | 5.21(0.05-30.3) | 0.86 | 1.13 | 6.15 | 8.28 | 4.03(1.06-11.4) |
| 2-Methylglyceric acid | 1.43 | 1.03 | 1.23 | 4.80 | 1.68(0.31-5.54) | 0.77 | 0.69 | 1.89 | 4.78 | 2.26(0.23-13.5) | 1.50 | 1.23 | 2.27 | 5.87 | 2.71(0.10-10.8) | 1.15 | 1.48 | 2.44 | 5.10 | 2.54(0.15-13.7) | 1.28 | 1.25 | 1.63 | 3.90 | 1.99(0.71-3.04) |
| 2-Methylthreitol | 0.98 | 1.76 | 6.12 | 18.8 | 4.98(0.26-25.6) | 1.12 | 1.60 | 7.80 | 6.91 | 4.87(0.34-17.1) | 0.87 | 1.22 | 7.25 | 6.91 | 4.06(0.25-12.9) | 1.10 | 1.33 | 5.93 | 7.78 | 4.03(0.26-15.4) | 1.00 | 2.01 | 6.60 | 7.24 | 4.17(1.93-7.93) |
| 2-Methylerythritol | 1.85 | 3.46 | 14.2 | 37.2 | 10.4(0.36-49.2) | 2.44 | 3.05 | 19.9 | 15.8 | 11.6(0.75-48.9) | 2.07 | 2.62 | 19.4 | 14.1 | 9.54(0.70-30.0) | 2.44 | 2.81 | 18.0 | 16.0 | 9.79(0.59-31.8) | 2.14 | 4.32 | 17.8 | 15.5 | 10.0(4.79-19.1) |
| Sum of $SOA_I$ tracers | 5.41 | 7.50 | 26.9 | 89.6 | 23.0(1.45-113) | 5.13 | 5.87 | 36.7 | 32.7 | 22.6(1.54-93.8) | 5.74 | 6.30 | 36.1 | 39.8 | 21.9(1.38-77.3) | 5.68 | 5.95 | 33.9 | 51.2 | 24.1(1.19-103.) | 5.82 | 9.46 | 36.9 | 40.8 | 23.0(10.8-49.3) |
| β-Caryophyllenic acid | 9.17 | 3.84 | 2.96 | 5.17 | 5.20(1.77-11.8) | 3.71 | 2.33 | 2.33 | 2.30 | 2.64(0.45-8.47) | 6.73 | 2.38 | 2.99 | 5.17 | 4.31(1.45-10.3) | 5.49 | 0.11 | 2.88 | 4.79 | 3.31(nd-11.4) | 7.49 | 2.39 | 2.73 | 3.93 | 3.84(1.82-7.07) |
| $SOA_M$ (µg m$^{-3}$) | 1.13 | 1.01 | 1.08 | 2.84 | 1.30(0.35-3.05) | 0.67 | 0.81 | 1.29 | 1.12 | 1.01(0.17-3.73) | 0.80 | 0.88 | 1.37 | 1.72 | 1.19(0.28-3.79) | 0.95 | 0.38 | 1.10 | 1.93 | 1.08(0.10-3.56) | 0.99 | 0.72 | 1.18 | 1.57 | 1.07(0.60-1.30) |
| $SOA_I$ (µg m$^{-3}$) | 0.07 | 0.10 | 0.34 | 0.97 | 0.27(0.01-1.25) | 0.07 | 0.08 | 0.44 | | 0.29(0.02-1.11) | 0.07 | 0.08 | 0.46 | 0.43 | 0.25(0.01-0.82) | 0.07 | 0.09 | 0.42 | 0.46 | 0.25(0.01-0.85) | 0.07 | 0.12 | 0.41 | 0.42 | 0.25(0.11-0.47) |
| $SOA_C$ (µg m$^{-3}$) | 0.84 | 0.35 | 0.27 | 0.47 | 0.47(0.16-1.08) | 0.34 | 0.21 | 0.21 | 0.21 | 0.24(0.04-0.77) | 0.62 | 0.22 | 0.27 | 0.47 | 0.39(0.13-0.94) | 0.50 | 0.01 | 0.26 | 0.44 | 0.30(nd-1.05) | 0.69 | 0.22 | 0.25 | 0.36 | 0.35(0.16-0.64) |
| BSOA (µg m$^{-3}$) | 2.04 | 1.46 | 1.70 | 4.28 | 2.05(0.53-4.33) | 1.08 | 1.11 | 1.98 | 1.76 | 1.55(0.26-5.35) | 1.49 | 1.18 | 2.11 | 2.62 | 1.84(0.56-5.43) | 1.52 | 0.47 | 1.79 | 2.82 | 1.65(0.16-5.24) | 1.74 | 1.06 | 1.84 | 2.35 | 1.68(0.96-2.19) |

"na" means not available and "nd" means not detected.

Table S2 SOA tracers and $f_{SOA}$ and $f_{SOC}$ values for SOA estimation

| | Monoterpenes [a] | Isoprene [b] | β-Caryophyllene [b] |
|---|---|---|---|
| SOA Tracers [c] | PNA (15%) [d] | 2-MTLs (41%) [d] | CA (157%) [d] |
| | PA (34%) [d] | 2-MGA (43%) [d] | |
| | MBTCA (62%) [d] | 3-MeTHF-3,4-diols (52%) | |
| | HGA (96%) [d] | $C_5$-alken triols (93%) | |
| | HDMGA (67%) [d] | | |
| $f_{SOA}$ (µg µg$^{-1}$) | 0.044 (48%) [e] | 0.063 (25%) [e] | 0.0109 (22%) [e] |
| $f_{SOC}$ (µg µgC$^{-1}$) | 0.059 | 0.155 | 0.023 |

[a] The $f_{SOA}$ and $f_{SOC}$ values for monoterpenes are calculated based on the data reported by Offenberg et al. (2007). [b] The $f_{SOA}$ and $f_{SOC}$ values for isoprene, and β-caryophyllene are reported by Kleindienst et al. (2007). [c] The numbers in brackets are uncertainties in tracer measurement. [d] These tracers are used to calculate $f_{SOA}$ and estimate ambient SOA. [e] The numbers in brackets are the uncertainties of $f_{SOA}$ values reported by Kleindienst et al. (2007).

Table S3 Correlation analysis of HO$_2$-channel SOA$_I$ tracers with O$_3$

| | Coefficient (r) | $p$-value |
|---|---|---|
| 3-MeTHF-3,4-diols | 0.343 | <0.001 |
| $C_5$-alkene triols | 0.388 | <0.001 |
| 2-Methyltetrols | 0.386 | <0.001 |
| HO$_2$-chanle SOA$_I$ tracers | 0.409 | <0.001 |

Table S4 Correlations among HO$_2$-channel SOA$_I$ tracers

| | 3-MeTHF-3,4-diols | $C_5$-alkene triols | 2-Methyltetrols |
|---|---|---|---|
| 3-MeTHF-3,4-diols | 1 | 0.789 | 0.792 |
| $C_5$-alkene triols | | 1 | 0.787 |
| 2-Methyltetrols | | | 1 |

All the correlations are significant (p<0.001)

Table S5 Rate constants and lifetimes of SOA precursors

| | **α-Pinene** | **β-Pinene** | **Isoprene** | **β-Caryophyllene** |
|---|---|---|---|---|
| | Rate constants at 298 K (cm$^3$ molecules$^{-1}$ s$^{-1}$) [a] | | | |
| **OH** | $5.25 \times 10^{-11}$ | $7.88 \times 10^{-11}$ | $9.99 \times 10^{-11}$ | $1.97 \times 10^{-10}$ |
| **O$_3$** | $9.01 \times 10^{-17}$ | $1.50 \times 10^{-17}$ | $1.28 \times 10^{-17}$ | $1.16 \times 10^{-14}$ |
| | Lifetimes (hrs) [b] | | | |
| **OH** | 0.53 | 0.35 | 0.28 | 0.14 |
| **O$_3$** | 3.64 | 21.9 | 25.7 | 0.03 |

[a] Rate constants are provided by MCMv3.2 (http://mcm.leeds.ac.uk/MCMv3.2).
[b] Lifetimes are estimated using summer average concentration of OH radical (~$1 \times 10^7$ molecules cm$^{-3}$) in the PRD (Hofzumahaus et al., 2009), and annual average O$_3$ concentration (67.7 µg m$^{-3}$) in Table S1.

Table S6 Concentrations of isoprene SOA products at HS and TS sites

| | HS 20150701 | TS 20150701 |
|---|---|---|
| 2-Methyltetrol sulfates (ng m$^{-3}$) | 6.65 | 2.99 |
| $C_5$-alkene triols (ng m$^{-3}$) | 11.5 | 10.8 |
| 2-Methyltetrols (ng m$^{-3}$) | 41.8 | 31.2 |
| 3-MeTHF-3,4-diols (ng m$^{-3}$) | 0.482 | 0.227 |

Table S7 Correlations of BSOA with sulfate and $O_x$ during fall-winter in 2008 at WQS

| | Sulfate (2008-WQS) | | | $O_x$ (2008-WQS) | |
|---|---|---|---|---|---|
| | Slope | *p*-value | % [a] | Slope | *p*-value |
| $SOA_M$ | 0.023 | 0.005 | 50 | - | 0.551 |
| $SOA_I$ | 0.032 | <0.001 | 76 | - | 0.509 |
| $SOA_C$ | 0.032 | <0.001 | 87 | - | 0.139 |
| BSOA | 0.087 | <0.001 | 69 | - | 0.563 |

[a] Percentages of SOA reduction at 50% decline of sulfate or $O_x$.

**References:**

[revised manuscript text omitted]